# Targeted protein degradation in mycobacteria uncovers antibacterial effects and potentiates antibiotic efficacy

Harim I. Won [1,4], Samuel Zinga [1,4], Olga Kandror[1], Tatos Akopian[1], Ian D. Wolf[1], Jessica T. P. Schweber[1], Ernst W. Schmid[2], Michael C. Chao [1], Maya Waldor[1], Eric J. Rubin [1] ✉ & Junhao Zhu[1,3] ✉

Proteolysis-targeting chimeras (PROTACs) represent a new therapeutic modality involving selectively directing disease-causing proteins for degradation through proteolytic systems. Our ability to exploit targeted protein degradation (TPD) for antibiotic development remains nascent due to our limited understanding of which bacterial proteins are amenable to a TPD strategy. Here, we use a genetic system to model chemically-induced proximity and degradation to screen essential proteins in *Mycobacterium smegmatis* (*Msm*), a model for the human pathogen *M. tuberculosis* (*Mtb*). By integrating experimental screening of 72 protein candidates and machine learning, we find that drug-induced proximity to the bacterial ClpC1P1P2 proteolytic complex leads to the degradation of many endogenous proteins, especially those with disordered termini. Additionally, TPD of essential *Msm* proteins inhibits bacterial growth and potentiates the effects of existing antimicrobial compounds. Together, our results provide biological principles to select and evaluate attractive targets for future *Mtb* PROTAC development, as both standalone antibiotics and potentiators of existing antibiotic efficacy.

Antibiotics, like many small molecule therapeutics, traditionally exert their mechanism of action through modulating a specific molecular target, whether by directly binding to an enzymatic active site or inducing allosteric conformational changes. Efforts to develop new antibiotics largely apply this same paradigm, but development remains challenging due to the high failure rate[1] and high cost[2]. As a result, the discovery of new antibiotics has stalled in recent decades, including of those to treat bacterial infections like tuberculosis (TB), which remains one of the world's leading infectious killers. The continued use of old antibiotics has compounded this problem, by giving rise to multi-drug resistant TB strains which are incredibly difficult to treat[3]. Because the traditional approach of single target modulation has slowed in its ability to discover powerful new anti-

tubercular agents, we sought to apply a novel modality to TB antibiotic development.

Targeted protein degradation (TPD) is an emerging therapeutic modality that has progressed from concept[4] to Phase 3 clinical trials[5] within the past two decades. The major class of molecules that enable protein level regulation through TPD are the proteolysis-targeting chimeras (PROTACs). These molecules are heterobifunctional, comprising two small molecule ligands joined by a chemical linker. One of the ligands binds an E3 ubiquitin ligase while the other binds a given target protein. The PROTAC induces the proximity of both the target protein and the E3 ligase, resulting in polyubiquitination of the target and its subsequent degradation by the eukaryotic proteasome. Other TPD approaches employ the lysosomal[6] and autophagic[7,8] degradation

[1]Department of Immunology and Infectious Diseases, Harvard T.H. Chan School of Public Health, Boston, MA 02115, USA. [2]Department of Biological Chemistry and Molecular Pharmacology, Harvard Medical School, Blavatnik Institute, Boston, MA 02115, USA. [3]CAS Key Laboratory of Pathogen Microbiology and Immunology, Institute of Microbiology, Chinese Academy of Sciences, Beijing, China. [4]These authors contributed equally: Harim I. Won, Samuel Zinga. ✉e-mail: erubin@hsph.harvard.edu; zhujh@im.ac.cn

machinery. Despite its great potential, the development of TPD-based antimicrobials remains largely conceptual. Rather than using host degradation machinery in a TPD strategy for bacteria, induced autoproteolysis would instead deliver proteins essential for bacterial viability to bacterial degradation machinery, resulting in their degradation and subsequent cell death or growth inhibition. In mycobacteria, options for degradation machinery include the mycobacterial proteasome, which uses the PafA ligase (a system analogous to the ubiquitination system employed by PROTACs[9]), or the Clp proteolytic complexes (i.e., ClpC1P1P2 and ClpXP1P2, which are essential in mycobacteria[10]).

Recently, the first TPD proof-of-concept in bacteria demonstrated that heterobifunctional bacterial PROTACs (BacPROTACs) could mediate the inducible degradation of target proteins in *Mycobacterium smegmatis (Msm)* cells[11,12]. This approach relied on direct delivery of target proteins to the multicomponent ClpC1P1P2 proteolytic complex (Fig. 1a), resulting in their degradation. In this work, BacPROTAC-mediated degradation of different targets was shown to induce sensitivity to D-cycloserine or auxotrophy for L-threonine, highlighting the potential of TPD as the foundation of a new class of antibiotic degraders[11]. The same group later reported that dimeric-Cyclomarin A analog could kill *M. tuberculosis (Mtb)* by inducing auto-degradation of ClpC1 and other ClpC homologs in mycobacteria[12]. However, it remains unclear whether all proteins are equally viable targets for this approach and for future drug development.

In this work, we develop a rational platform for TPD target selection in mycobacteria and survey a set of diverse, essential mycobacterial proteins to examine their suitability for a TPD approach. Our genetic platform enables the rapid assessment of different target proteins for TPD, the output of which can subsequently train machine learning models to uncover critical protein features which underlie TPD potential. We also find that targeted degradation of particular substrates is sufficient to cause cellular growth inhibition and to potentiate killing by clinical and pre-clinical TB antibiotics even when targeted degradation alone functions at sub-inhibitory levels.

## Results

### A chemical genetic platform for evaluating targeted protein degradation in mycobacteria

The development of a targeted protein degradation (TPD) strategy for *Mtb* infection necessitates the identification of protein targets that are required for bacterial growth and are amenable to proximity mediated degradation by bacterial proteolytic systems. However, the systematic evaluation of vulnerable protein targets for TPD drug development is infeasible using a chemical-forward approach (i.e., identifying chemical ligands to over 400 genes that are essential for *Mtb* growth[10]). Therefore, we sought to develop a platform for rational target selection, which would enable the prioritization of targets before chemical screening for binding ligands. To accomplish this, we needed a system that would enable regulated, inducible proximity of native proteins and a mycobacterial proteolytic complex like ClpC1P1P2, which has recently been used for TPD in bacteria[11,12]. For this purpose, we chose to create genetic fusions with FRB, the rapamycin-binding domain of

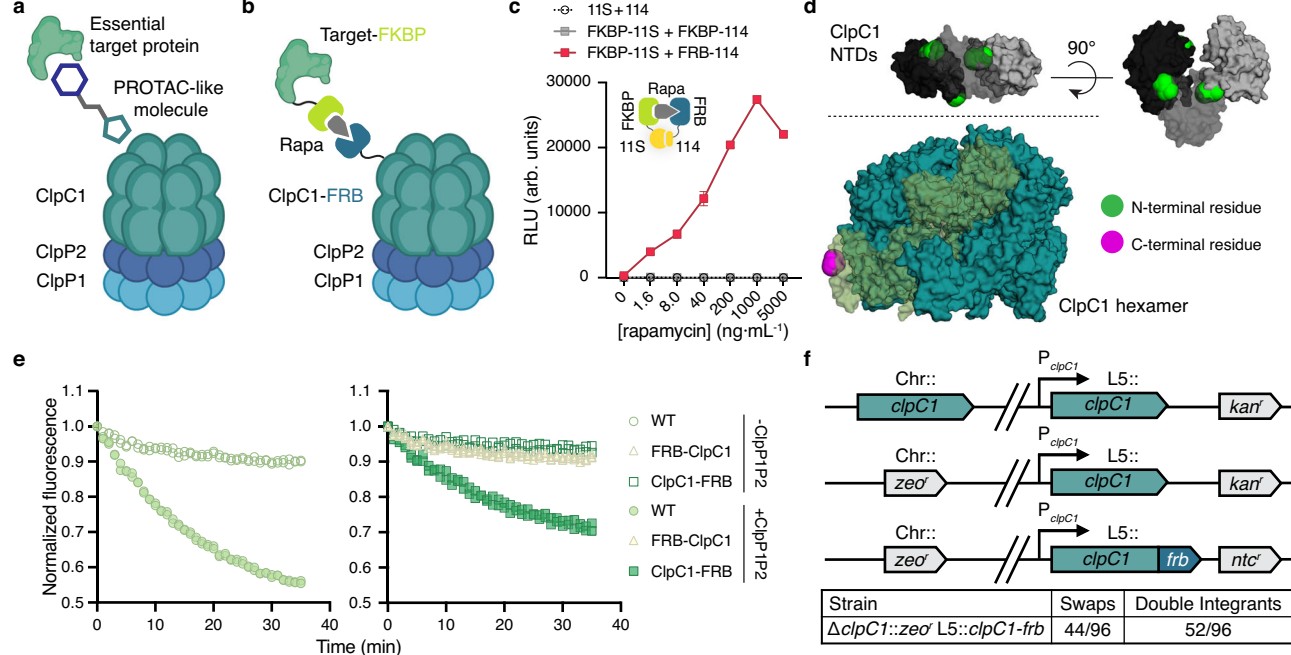

**Fig. 1 | The FRB-FKBP dimerizable domains can be used to induce proximity in mycobacteria. a** The concept of bacterial targeted protein degradation (TPD), in which an essential target protein is delivered to the ClpC1P1P2 proteolytic complex by a heterobifunctional molecule. **b** Schematic of our chemical-genetic approach to induced proximity, in which fusions to the FRB and FKBP domains are dimerized with the addition of rapamycin. **c** Luminescence in live cells expressing split NanoLuc (Nluc) fragments with and without fusion to FRB and FKBP measured in relative light units (RLU). Density matched log phase cells incubated with the luciferase substrate furimazine and a range of rapamycin concentrations at 37 °C for 10 min. **d** Crystal structure of the stabilized mutant *Mtb*ClpC1 hexamer (teal, PDB: 8A8U) with highlighted monomer (translucent green) and three visible N-terminal domains (NTDs) which cannot be assigned to specific protomers due to invisibility of the linker region (grayscale, PDB: 6PBS). ClpC1 N- and C-terminal residues highlighted in green and magenta, respectively. **e** Fluorescence of eGFP-

ssrA measuring in vitro protease activity of ClpC1P1P2 complex with WT *Mtb*ClpC1 (left) or FRB-ClpC1 and ClpC1-FRB (right). Purified *Mtb*Clp proteins incubated with eGFP-ssrA substrate at 37 °C. **f** (top) Schematic of the *clpC1* L5 allele swap, in which a second copy of *clpC1* is integrated at the L5 phage *attB* site. This enables recombineering-mediated knockout of chromosomal (Chr::) *clpC1*, followed by an L5 integrase-mediated swap for the *clpC1-frb* allele and an alternative resistance marker. (bottom) Quantification of the *clpC1-frb* swap; true swaps carry only the second resistance marker, whereas double integrants carry both (and both *clpC1* alleles). For (**c**), data are mean ± s.d. of three technical replicates and are representative of two independent experiments; **e**, data are individually plotted technical replicate measurements, normalized to time = 0 h, and are representative of two independent experiments. **a**, **b** Created with BioRender.com, released under a Creative Commons Attribution-NonCommercial-NoDerivs 4.0 International license. Source data are provided with this paper.

mTOR, and FKBP12, which binds both rapamycin and FK506[13]; FRB and FKBP are small (~12 kDa) proteins which dimerize with nanomolar affinity only in the presence of rapamycin, forming a ternary complex[14,15]. We fused ClpC1 with the FRB domain and candidate proteins with FKBP, creating strains where the presence of rapamycin induces the proximity of target proteins to ClpC1 (Fig. 1b). To enable rapid screening of targets which were most effectively degraded, we built this system in the non-pathogenic mycobacterial model species *Msm*.

Though FRB and FKBP have been widely used to conditionally dimerize proteins in mammalian cells[15] and, more recently, in *Escherichia coli*[16], they had not yet been evaluated in live mycobacterial cells. Therefore, we first assessed whether rapamycin-dependent dimerization occurred in *Msm* cells by fusing each domain to the NanoLuc (Nluc) 11S (large) fragment and 114 (small) peptide split reporter system[17]. Compared to other split reporter systems (e.g., fluorescent proteins, β-galactosidase), the split Nluc (NanoBiT) system features the rapid signal detection, sensitivity, and dynamic range of luciferase assays while also being optimized to have minimal affinity between the split components. We expressed the FRB and FKBP Nluc fusion proteins in *Msm* and observed rapamycin-dependent luciferase activity only when both rapamycin-binding domains were fused to the split Nluc fragments (i.e., FRB-11s and FKBP-114), and this activity appeared to saturate at a rapamycin concentration of 1 μg·mL$^{-1}$ (Fig. 1c). We also observed the characteristic hook effect resulting from the formation of unproductive binary complexes at high concentrations rather than necessary ternary complexes required for induced proximity[18,19]. Importantly, all concentrations of rapamycin we used to induce proximity in this work were inert with respect to bacterial growth and were orders of magnitude below the highest tested concentration which itself also did not inhibit bacterial growth in wild-type mc²155 strain *Msm* cells (Supplementary Fig. 1).

To generate the ClpC1 and FRB fusion protein, the FRB domain could be fused to either the N-terminal or C-terminal end. Based on the crystal structures of the *Staphylococcus aureus* ClpC and *Mtb* ClpC1 hexamer[20,21], we reasoned while N-terminal fusion might facilitate the direct delivery of target proteins to the ClpC1 hexamer pore, the genetic engineering of the ClpC1 C-terminus would be better tolerated by cells given the degree to which the N-terminal ends were nestled among the ClpC1 N-terminal domains (NTDs) compared to the relative accessibility of the C-terminal end (Fig. 1d). To compare these two scenarios, we purified both the N-terminal and C-terminal *Mtb*ClpC1 fusions (FRB-ClpC1 and ClpC1-FRB, respectively; Supplementary Fig. 2a) to measure each fusion protein's ability to degrade enhanced green fluorescent protein tagged with ssrA (eGFP-ssrA). This target was selected as WT *Mtb*ClpC1 in association with ClpP1P2 had been previously shown to be able to degrade eGFP-ssrA in vitro[22,23]. Indeed, we found FRB-ClpC1 had no measurable protease activity, while ClpC1-FRB was able to degrade eGFP-ssrA, albeit to a lesser degree than WT ClpC1 (Fig. 1e). We also observed this trend among the different ClpC1 fusion variants without ClpP1P2 when measuring ATPase activity, though here, FRB-ClpC1 showed some minimal activity (Supplementary Fig. 2b, c).

To generate a strain carrying a single copy of *clpC1-frb* under the native *clpC1* promoter, we replaced wild-type *clpC1* with a *clpC1-frb* allele via an L5 integrase allele swap[24] (Fig. 1f, top). We generated viable *Msm* strains with clean swaps for *clpC1-frb* alleles at a rate of 45.8%, successfully verifying that ClpC1-FRB remains functional in vivo and can complement wild-type ClpC1's essential role in normal mycobacterial growth (Fig. 1f, bottom). We further confirmed that the addition of the FRB domain did not disrupt cellular physiology by measuring growth and found that the strain expressing ClpC1-FRB shared growth characteristics with parental *Msm* (Supplementary Fig. 3). This strain served as the basis for our platform to assess TPD in mycobacteria and examine whether we could mediate the degradation of specific target proteins by induced proximity to ClpC1.

## Rapamycin-induced proximity to ClpC1 is sufficient to degrade a native mycobacterial protein

To test whether rapamycin-induced proximity to ClpC1 was sufficient for mycobacterial protein degradation, we constructed target protein fusions with FKBP and eGFP, introduced them as merodiploid copies into the Tweety phage *attB* site[25] in both a WT *clpC1* and Δ*clpC1* L5::*clpC1-frb* background (Fig. 2a, b), and measured the effect of rapamycin on fluorescence in *Msm*. Before testing different native targets, we assessed whether the minimal backbone, FKBP-eGFP alone, could be directed for degradation by ClpC1 or if degradation was more substrate selective. We found that heterologously expressed FKBP-eGFP was not degraded regardless of *clpC1* background (Fig. 2c), suggesting either that it is a poor ClpC1 substrate (perhaps due to eGFP's characteristic, protease-resistant tight folding[26]) or that the geometry of delivery we achieved is incompatible with efficient degradation. To test whether the absence of rapamycin-induced degradation was due to a possible cryptic obstruction of the rapamycin-mediated protein-protein interaction, we examined the subcellular distribution of the FKBP-eGFP signal with or without rapamycin treatment. Previous work indicated that full-length ClpC1-eGFP locates near the plasma membrane and enriches at the subpolar region in a non-homogeneous manner[27] which we also observed (Supplementary Fig. 4a, b). While FKBP-eGFP in the *clpC1-frb* background diffuses in the cytosol, the addition of rapamycin results in re-localization of FKBP-eGFP to the subpolar membrane compartment (Supplementary Fig. 4c), indicating that rapamycin successfully mediates the physical interaction between FKBP-eGFP and ClpC1-FRB and further suggesting that FKBP-GFP alone is a suboptimal substrate for ClpC1.

We next selected RpoA, a small (38 kDa) and essential component of the RNA polymerase (RNAP) holoenzyme, to test whether the addition of an endogenous protein could potentiate proximity-induced degradation by ClpC1. Indeed, we found that rapamycin is sufficient to direct RpoA-FKBP-eGFP for degradation, with substantial loss of eGFP signal only in the *clpC1-frb* background (Fig. 2d). To validate that the observed loss of eGFP signal was the result of bona fide protein degradation, we showed a rapamycin-dependent reduction in protein levels of the full length RpoA-FKBP-eGFP fusion protein only in the *clpC1-frb* background (Fig. 2e). Furthermore, to assess the dynamics of this targeted protein degradation of RpoA, we examined individual cells by time-lapse fluorescence microscopy and observed a rapamycin-dependent reduction of eGFP signal only in the *clpC1-frb* background within 8 h of induction (Fig. 2f, g). Our findings demonstrate that a genetically encoded system for chemically-induced proximity to ClpC1 which is orthogonal to the chemical approach developed by ref. 11., is sufficient to degrade a native mycobacterial protein.

## Mycobacterial proteins are differentially susceptible to ClpC1-mediated TPD

To identify suitable endogenous targets for TPD more systematically, we sought to fuse a set of native mycobacterial proteins to FKBP-eGFP and to assess their degradation potential (Fig. 3a and Supplementary Fig. 6a). The fluorescent degradation kinetics of RpoA as measured by flow cytometry (Fig. 3b) correlated well with the signal observed by time-lapse microscopy (Supplementary Fig. 6b), motivating our use of flow cytometry as the basis for our target screening assay. To increase the generalizability of our screen in *Msm* to *Mtb*, we focused on protein orthologs which are highly conserved between the two species and vulnerable to depletion in both[10,28,29]. In total, we identified 348 conserved *Msm* proteins that met these criteria (Methods, Supplementary Data 1), from which we chose 54 proteins with diverse functional and biochemical properties (Supplementary Fig. 7a, b). Using high-throughput flow cytometry, we quantified the effect of rapamycin on the fluorescence of these

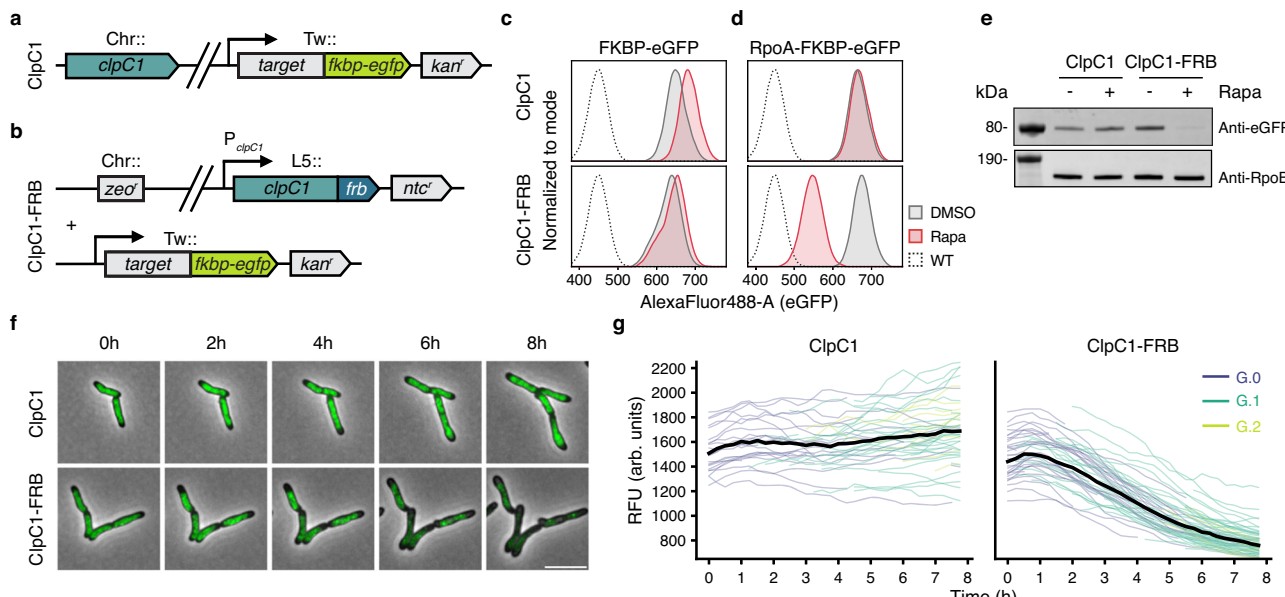

**Fig. 2 | Rapamycin redirects a native mycobacterial protein for degradation by ClpC1-FRB. a, b** Schematics of reporter strains generated to assess target degradation, with *substrate-fkbp-egfp* genes delivered to the Tweety (Tw::) phage *attB* site in a WT *clpC1* (**a**) or Δ*clpC1* L5::*clpC1-frb* (**b**) background. **c, d** Fluorescence of live cells as a proxy for protein levels of FKBP-eGFP (**c**) or RpoA-FKBP-eGFP (**d**) in the WT *clpC1* (top) or *clpC1-frb* (bottom) background. Density matched log phase cells incubated with DMSO or 0.1 μg·mL⁻¹ rapamycin with shaking at 37 °C for 24 h. **e** Western blot analysis of RpoA-FKBP-eGFP with DMSO or 0.1 μg·mL⁻¹ rapamycin addition in both strain backgrounds. Density matched log phase cells incubated with DMSO or 0.1 μg·mL⁻¹ rapamycin with shaking at 37 °C for 24 h. **f** Live cell, wide-field fluorescence microscopy time-lapse images of cells expressing RpoA-FKBP-

eGFP with 0.1 μg·mL⁻¹ rapamycin in both strain backgrounds. Scale bar, 5 μm. **g** Quantitation of all captured fields during time-lapse in (**f**), measuring median fluorescent signal in relative fluorescence units (RFU) in the FITC channel across cells over time. Colored lines represent individual RFUs (ClpC1, *n* = 77; ClpC1-FRB, *n* = 105) that were initially plated (G.0, purple), first-generation (G.1, teal), or second-generation (G.2, lime green). For (**c, d**), data are representative of three independent experiments; **e** data represent one independent experiment. **f, g** data are representative images selected from among 4 fields for each and are representative of two independent experiments. Source data are provided with this paper.

targets over time and found that most of the tested native proteins were able to be targeted for degradation by rapamycin (Fig. 3c and Supplementary Fig. 7c) and that the observed fluorescent signal decay could be modeled by an exponential decay function (Supplementary Fig. 7d), allowing us to calculate a rate constant for each target (Fig. 3d and Supplementary Data 2). While some targets displayed slower degradation kinetics – such as PyrG, which did not reach maximal, steady state degradation until 12 h (*dG/dt* = 0.026) – we found that others, such as the transcription factor RbpA, were rapidly degraded to a steady state baseline by 3 h (*dG/dt* = 1.414; Fig. 3d and Supplementary Fig. 7d).

Interestingly, although all targets were expressed from the same constitutive promoter, we observed variance in the steady state fluorescence among the strains in the absence of rapamycin, indicating that there may be some basal post-translational regulation of some targets, possibly through proteolysis. We wondered whether these actively degraded proteins could also be more susceptible to TPD, if they are naturally recognized by endogenous proteases. Indeed, we found an inverse correlation with the $\log_2$ degradation rate of each target and its steady state fluorescence (Fig. 3e; Pearson's $r = -0.69$, $P = 6.59 \times 10^{-9}$). We also leveraged previously published transcriptomic[30] and quantitative proteomic[31] datasets of the *Mtb* H37Rv lab strain to estimate the steady-state protein/RNA ratios of *Mtb* genes. We found that this orthogonal metric, which roughly reflected the composite effect of translation efficiency and protein turnover, also exhibited a mild, though statistically insignificant, negative correlation with the $\log_2$-transformed degradation rate of the corresponding *Msm* homologs (Supplementary Fig. 8a). Together, these results imply that this intriguing relationship between steady-state protein abundance and protein degradability is probably not confounded by unknown technical caveats but is rather a reflection of intrinsic biological differences among these proteins.

To assess the effect of rapamycin-induced proximity to ClpC1 on protein localization and protein levels, we imaged each strain using fluorescence microscopy and again observed high concordance between the flow cytometry and microscopy datasets (Fig. 3f and Supplementary Fig. 8b, c). We also confirmed the gradient of degradation potential of the tested targets (Supplementary Fig. 8c). Notably, for several substrates, including NrdE, SecA1, and InhA, the addition of rapamycin resulted in profound re-localization of fluorescent signal (Fig. 3f, Supplementary Fig. 8c). Though these observations could reflect biology, these punctate foci could alternatively form due to oligomerization of eGFP due to increases in local concentrations near ClpC1-FRB. Finally, we examined whether rapamycin is stable in *Msm* culture conditions and found that eGFP levels remain low for at least 48 h post-inoculation, indicating that rapamycin is not rapidly metabolized below effective levels in that time period (Supplementary Fig. 9).

**Protein sequence and structural features underlie TPD potential**
The 54 substrates depicted in Fig. 3e vary not only in their measured degradation potential, but also by other primary sequence features, such as isoelectric point and aliphatic index (Supplementary Fig. 7b). Therefore, we sought to leverage the diversity of this dataset to query the sequence and structural determinants underlying TPD potential. We assembled a diverse set of numeric protein descriptors, encompassing quantitative structure-activity relationship (QSAR) metrics[32,33], AlphaFold2 prediction confidence[34], protein disorder propensity[35], and additional metrics, and applied them to both the full-length proteins and their terminal segments (Methods, Supplementary Data 2). After feature selection, 40 of the 485 protein features describing the 54 substrates and their measured degradation constants (the training set) were used to train a Lasso regression model (Fig. 4a, Supplementary Data 2). To evaluate the trained linear model, we constructed

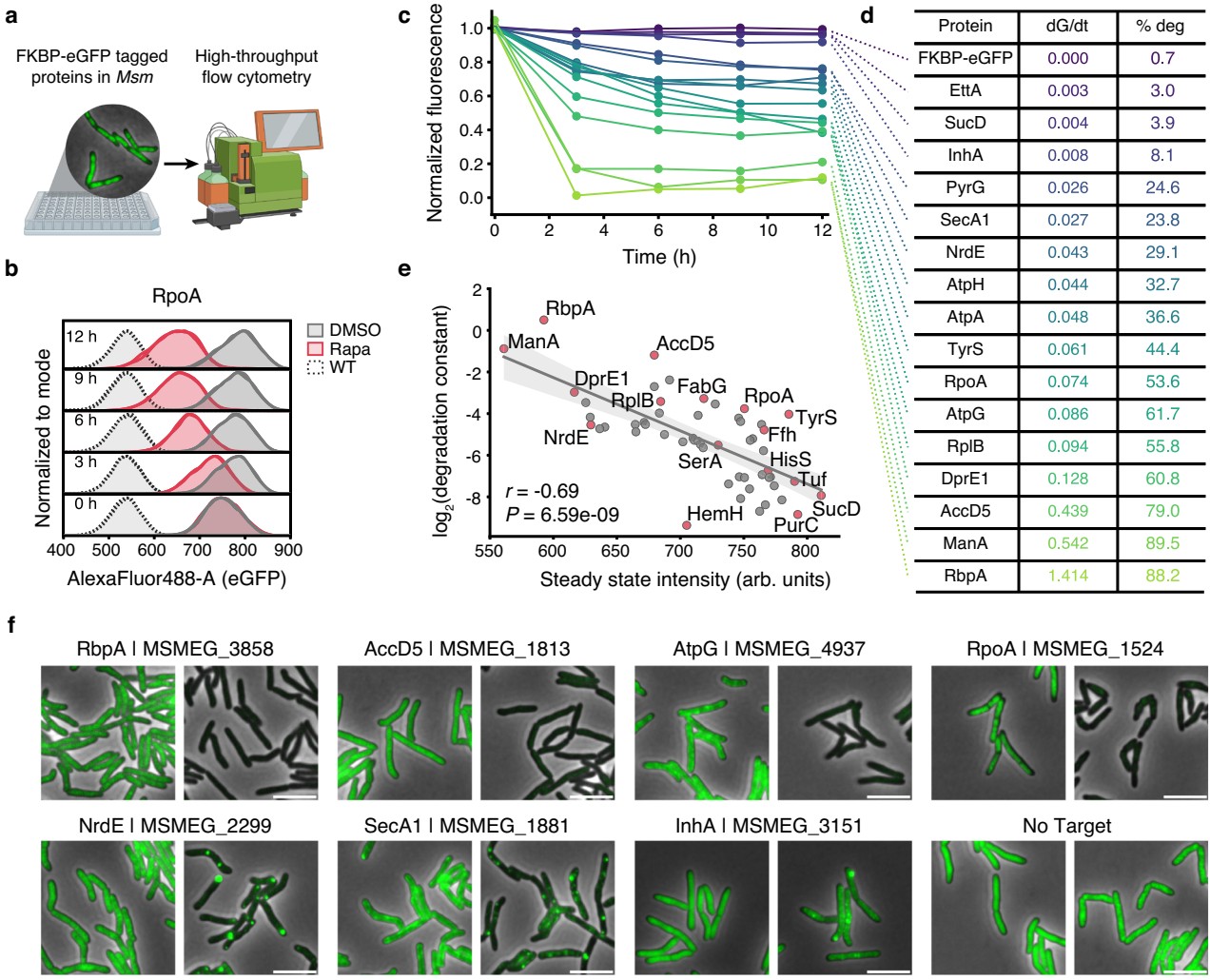

**Fig. 3 | Mycobacterial proteins are differentially susceptible to degradation by ClpC1-FRB. a** Schematic of high-throughput flow cytometry screen in *Msm* cells expressing targets tagged with FKBP-eGFP. **b** Fluorescence of live cells as a proxy for protein levels of RpoA-FKBP-eGFP in the *clpC1-frb* background over time. **c** Summary plot of selected target proteins; fluorescence of live cells as a proxy for protein levels in the *clpC1-frb* background over time. Density matched log phase cells incubated with DMSO or 0.1 µg·mL⁻¹ rapamycin with shaking at 37 °C for the indicated times. The flow cytometry fluorescence distribution and degradation kinetics of the full set of target proteins are included in Supplementary Figs. 7c, d. **d** Enumeration of selected target proteins and their respective rate constants (*dG/dt*) and proportion of total degradation (% deg). The corresponding statistics for the full set of 54 protein targets are included in Supplementary Data File 2. **e** Correlation plot relating the log₂-transformed degradation rate and the steady state intensity of the 54 target proteins listed in Supplementary Data File 2. **f** Live cell,

wide-field fluorescence microscopy images of cells expressing selected target proteins following 9 h of incubation with DMSO or 0.1 µg·mL⁻¹ rapamycin. Scale bar, 5 µm. For (**b**), data are two technical replicates, are representative of three independent experiments, and normalized to the mode; For (**c**), data are the mean of two technical replicates, are representative of two independent experiments, and normalized to fluorescent intensity of DMSO-treated cells at time = 0 h. The centerline in (**e**) denotes the linear fit between the measured steady state fluorescence intensity and the estimated degradation constant (log₂-transformed) and is bounded by the 95% confidence interval. The correlation coefficient (*r*) and the *P*-value in (**e**) were determined using a two-sided Pearson's correlation test. Data in (**f**) are representative microscopy images selected from among 12 fields for each strain and are representative of two independent experiments. **a** Created with BioRender.com, released under a Creative Commons Attribution-NonCommercial-NoDerivs 4.0 International license. Source data are provided with this paper.

additional reporter strains for 18 protein substrates and measured their degradation rate upon rapamycin treatment. We found high concordance between the predicted and the measured degradation rate of the additional 18 substrates (the validation set; Fig. 4a, Supplementary Fig. 10, Supplementary Data 2). The performance of the relatively simple linear model implied linear or near-linear relationships between a subset of sequence or structural features and protein degradation potential. Indeed, we found that many of the selected features exhibited a statistically significant correlation with protein degradation potential (Supplementary Fig. 11a). A key example is the mean of disorder propensity of the N-terminal 30 residues (Fig. 4b), which suggested that the more disordered the N-terminal sequence is, the more efficiently TPD could act on the protein. The predictive

power of this single numeric feature was further revealed by the terminal Disorder Propensity (fIDPnn) profiles of the 72 tested substrates which implied that the well-degraded proteins tended to have disordered N-termini, but not necessarily disordered C-termini (Fig. 4b, c). Notably, when we combined the two batches of protein substrates which were used for model training and validation and compared their measured degradation potential to the steady-state protein-to-RNA ratio of their *Mtb* homologs, we indeed observed a statistically significant inverse correlation (Supplementary Fig. 11b), which further supported our hypothesis that efficient TPD is reliant on the substrate's natural susceptibility to endogenous proteases. Having validated our linear prediction model, we applied it to predict the degradability of 348 essential proteins which are highly conserved

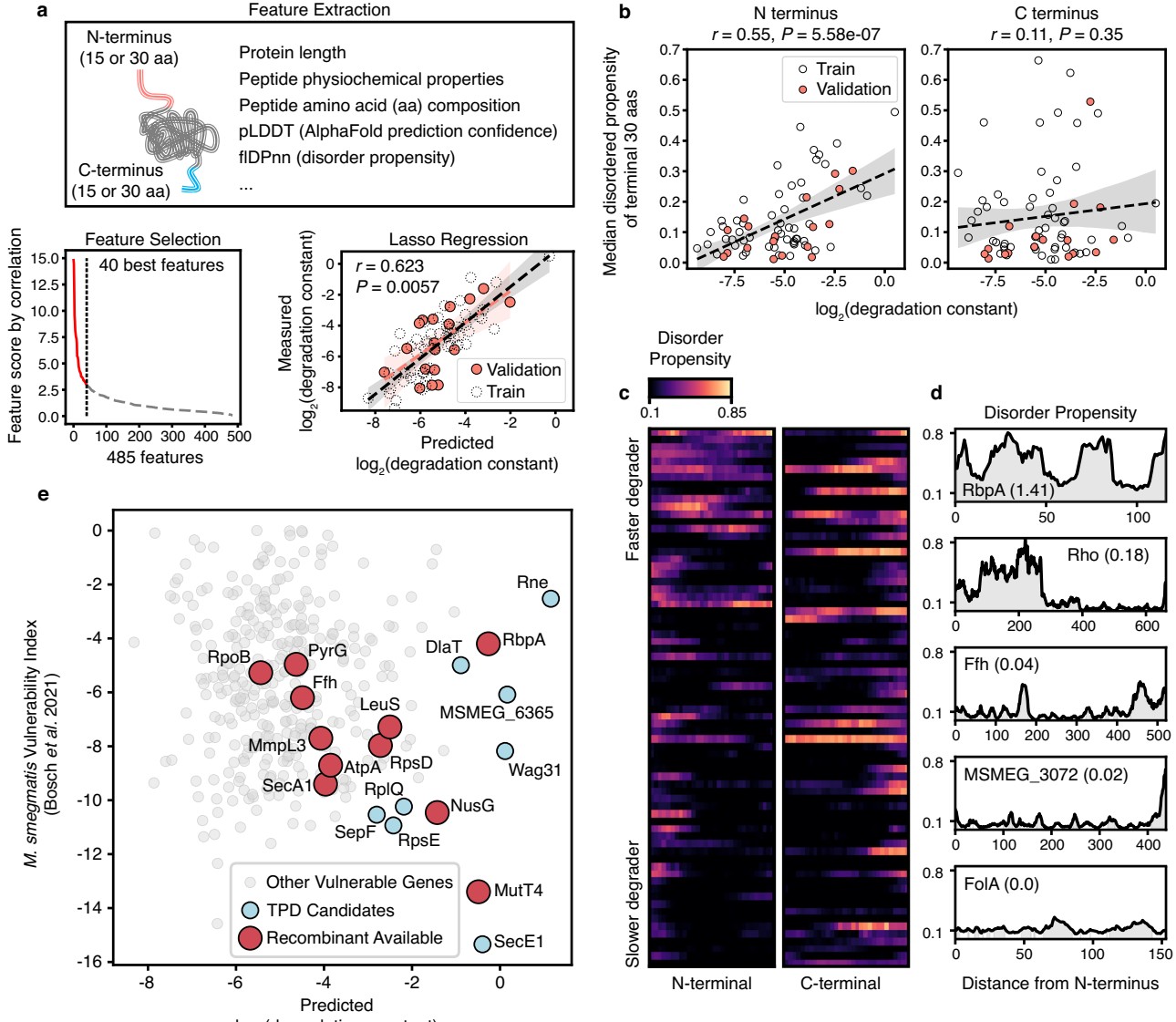

**Fig. 4 | Machine learning uncovers features underlying protein degradation potential. a** Schematic of our machine learning experiment encompassing protein feature extraction, feature selection, and Lasso regression. The accuracy of the Lasso regression model was estimated by calculating Pearson's correlation between the measured and the predicted $log_2$(degradation constant) of the 18 substrates used for model validation. $r$ and $P$ represent the coefficient and $P$-value of the Pearson's correlation analysis, respectively. **b**–**d** The disorder propensity of the N-termini, but not the C-termini of tested substrates positively associates with protein degradation potential. (**b**) depicts the linear association between the average disordered propensity of the first (N-termini) and the last (C-termini) 30 amino acids as measured by flDPnn method[35]. $r$ and $P$ represent the Pearson's coefficients and their $P$ values between the protein features and their measured $log_2$(degradation constant), respectively. (**c**) depicts the flDPnn predicted per-residue disorder propensity of the terminal 30 amino acids of each substrate. Here the terminal disorder propensity profiles of the 72 tested substrates (54 for model

training and 18 for validation) are stacked and ordered by their degradation constant (top to bottom). For both termini, the per-residue values are color-coded and ordered by their distance from the start codon (left to right). (**d**) demonstrates the full-length disorder propensity profiles of 5 representative proteins with varied degradation potential (degradation constant provided in parentheses). **e** Scatter plot depicting the predicted degradation potential of the 348 conserved essential *Msm* proteins along with their transcriptional vulnerability index as measured by a genome-wide CRISPRi screening[28]. Protein candidates of interest, including those predicted to be highly susceptible to TPD (light blue), or the ones for which we have generated chromosomally tagged TPD strains (red), are highlighted as colored dots. The centerlines (dashed black lines) in (**a**, **b**) denote the linear fit of the datapoints and are bounded by the 95% confidence interval. The correlation coefficients ($r$) and $P$-values in (**a**, **b**) were determined using a two-sided Pearson's correlation test. Source data are provided with this paper.

between *Mtb* and *Msm*, and identified TPD candidates which are also vulnerable to transcriptional perturbation via CRISPRi[28] (Fig. 4e, Supplementary Data 3).

## TPD of native mycobacterial proteins inhibits bacterial growth and potentiates existing antimycobacterial compounds

In our previous experiments revealing that induced proximity drives the degradation of several mycobacterial proteins, we did not expect

to see any impacts on growth because the strains we generated for these experiments were merodiploid for the protein being degraded (i.e., the genes were present in two copies), with only one of the copies tagged with *fkbp-egfp*. We therefore sought to test whether targeted degradation of native mycobacterial proteins predicted by our linear model to be highly susceptible to TPD (Fig. 4e) impacts bacterial growth. We accomplished this by endogenously tagging a subset of targets at their chromosomal loci with *fkbp-egfp* or *fkbp-flag* by

recombineering in a Δ*clpC1* L5::*clpC1-frb* background (Fig. 5a) and measuring the effect of rapamycin on bacterial growth. We selected targets spanning a range of predicted degradability and included known (e.g., RpoB), and potential (e.g., MmpL3, NusG[36] and RbpA[37]) drug targets which our model predicted to be efficiently degraded in our system (Fig. 4e). We observed that rapamycin-mediated degradation of NusG, RbpA, Ffh, MmpL3, and AtpA resulted in substantial growth delay in cells grown in liquid media, whereas the targeted degradation of RpoB, RpsD, MutT4, SecA1, PyrG, and LeuS had little to no effect on bacterial growth (Fig. 5b).

We then tested the extent to which endogenously tagged substrates could be degraded in a rapamycin-dependent fashion using fluorescence microscopy (Fig. 5c). It is worth noting that while most tested substrates were tagged with FKBP-eGFP, thereby permitting subsequent microscopy imaging, AtpA was excluded from this experiment as we found a cryptic duplication of the AtpA locus in the FKBP-eGFP tagged strain, but not in the FKBP-FLAG tagged counterpart (Supplementary Fig. 12a, b). Even though we expected the fluorescence of endogenous substrates to be confounded by variations in gene expression (Supplementary Fig. 12c), there still exists a robust positive correlation between the normalized fluorescence decrease of these FKBP-eGFP tagged endogenous substrates and their predicted degradation potential (Supplementary Fig. 12c–e)

In addition to changes in growth dynamics, fluorescence intensity, and protein localization (Fig. 5b, c), we also observed that rapamycin-induced degradation of several substrates, such as MmpL3, resulted in aberrant cell morphology that was reminiscent of reported morphological changes due to transcriptional perturbation[38] or chemical inhibition[39] (Fig. 5c, Supplementary Fig. 12f, g). Together, these results imply that our genetic platform effectively models chemically induced TPD of endogenous substrates, and that ClpC1-mediated degradation of certain substrates strongly inhibits mycobacterial growth.

The depletion of antibiotic targets, either at the protein or RNA level, has been shown to alter bacterial responses to compounds which inhibit those targets[40,41]. We therefore sought to test whether targeted degradation of protein complex components could sensitize bacteria to clinically-relevant antibiotics that target the components themselves or other members of the complexes they form (Fig. 5d). There was no sensitization to any antibiotic tested when FKBP-eGFP was used as the substrate (Supplementary Fig. 13a). By contrast, we observed over 10- and 20-fold decreases in the half-maximal minimum inhibitory concentration ($MIC_{50}$) of the RNA polymerase inhibitor rifampicin, when the polymerase $\beta$ subunit RpoB and the transcription facilitator RbpA were targeted for degradation[37,40], respectively (Fig. 5e). Fidaxomicin is another FDA-approved RNAP inhibitor that has regained attention for its excellent in vitro bactericidal activity against *Mtb*[42]. A recent study demonstrated that fidaxomicin jams mycobacterial RNAPs via direct interactions with the core complex and RbpA[43] and reported that removing the N-terminal tail of *Msm* RbpA weakened fidaxomicin binding as well as its growth inhibition activity. In support of the hypothesis that full-length RbpA is critical for maintaining the structural and functional integrity of mycobacterial RNAPs in the presence of both inhibitors, we found that the rapamycin induced degradation of RbpA indeed lowered the fidaxomicin $MIC_{50}$ by over four-fold (Fig. 5f). We also tested the impact of targeted degradation of RbpA on rifampicin-resistant (rif[R]) isolates and found that RbpA depletion could re-sensitize rif[R] *Msm* isolates to rifampicin, albeit to variable degrees depending on the specific mutation that conferred resistance (Supplementary Fig. 13b).

Compared to RpoB and RbpA, targeted degradation of MmpL3 and AtpA had a milder but still pronounced impact on *Msm*'s susceptibility to the MmpL3 inhibitor SQ109 and the ATP synthase inhibitor bedaquiline, respectively (Fig. 5g, h). Degrading RpoB, RbpA, and MmpL3 did not sensitize *Msm* to the antibiotics streptomycin and linezolid which inhibit translation by targeting the 30 S and 50 S

ribosomal subunits, respectively, but degrading AtpA resulted in mild sensitization to both antibiotics which could suggest that these cells were in a compromised state that is more susceptible to antibiotic stress more generally (Supplementary Fig. 13c, d) Because bedaquiline has been shown to elicit a much delayed bactericidal activity against *Mtb*[44,45], we additionally measured the survival fraction of *Msm* exposed to 10× the observed $MIC_{50}$ of bedaquiline in the presence or the absence of rapamycin-induced degradation of AtpA. We found that targeted degradation of AtpA substantially accelerated cell killing by bedaquiline compared to cells with no degradation, which did not demonstrate any killing at all (Fig. 5i, $P < 0.05$ for 27 and 48 h). In the absence of bedaquiline, rapamycin had no discernible effect on bacterial growth (Supplementary Fig. 13e). Taken together, our findings highlight current antibiotic targets as pathways which can also be attacked by ClpC1-mediated targeted protein degradation, yielding direct effects on inhibiting bacterial growth, but importantly also sensitizing and resensitizing bacteria to currently approved antibiotics and accelerating their killing kinetics.

## Discussion

In this work, we developed and validated a platform for the systematic identification of promising targets in the development of BacPROTAC targeted degrader antibiotics against mycobacteria. Genetic fusions to the FRB and FKBP domains enable the induced proximity of targets and the ClpC1 ATPase using the small molecule rapamycin, facilitating the subsequent proteolytic degradation of endogenous proteins. We show that many, but not all proteins we screened are effectively degraded using this approach, and that certain sequence properties, such as the disorder propensity of a protein's N-terminus, are predictors of degradation potential of mycobacterial proteins. Moreover, we demonstrate that TPD-mediated depletion of certain proteins represses bacterial growth and sensitizes *Msm* to antibiotics which target the same pathway. Our results highlight the potential of TPD as a new modality for antimicrobial therapeutics.

When selecting targets for a TPD strategy, access to the target surface by the degradation mediating machinery (e.g., E3 ligase for PROTACs) is essential. The geometry of the nascent ternary complex formed when targets are brought in proximity to degradation machinery is therefore a critical factor in determining whether proteins can be degraded or not[46]. In addition to the limitations on our ability to vary the geometry of potential targets, an additional layer of complexity to our ClpC1-mediated approach is that ClpC1 degradation has its own structural determinants. The rules governing whether ClpC1 will degrade a given protein are not well known. Previously, Lunge and others conducted quantitative proteomics to query the substrate pool of ClpC1 and reported that mycobacterial ClpC1 may preferentially degrade proteins with disordered termini[47]. Our study, which approached this question using an orthogonal method, provided additional evidence to their hypothesis. The lack of association between the disorder propensity of protein C-terminus and degradation potential is probably because in our system, the C-termini of protein substrates are fused to FKBP-eGFP and pulled away from the entry pocket of the ClpC1 unfoldase. While further structural evidence to fully elucidate ClpC1's substrate selectivity is still needed, our study along with previous ones may shed light on which native protein substrates should be prioritized for BacPROTAC drug design and screening.

Although we ultimately selected the ClpC1P1P2 proteolytic complex as the mediator of TPD in mycobacteria in our system, there are other endogenous protein degradation systems that may also be suitable[48]. The Pup proteasome system (PPS) is more analogous to PROTACs[49]; the mycobacterial PPS uses the pupylating ligase PafA, which is structurally distinct but functionally akin to human ubiquitin ligases, to mark endogenous proteins for degradation by the mycobacterial proteasome complex. In our proposed scheme of

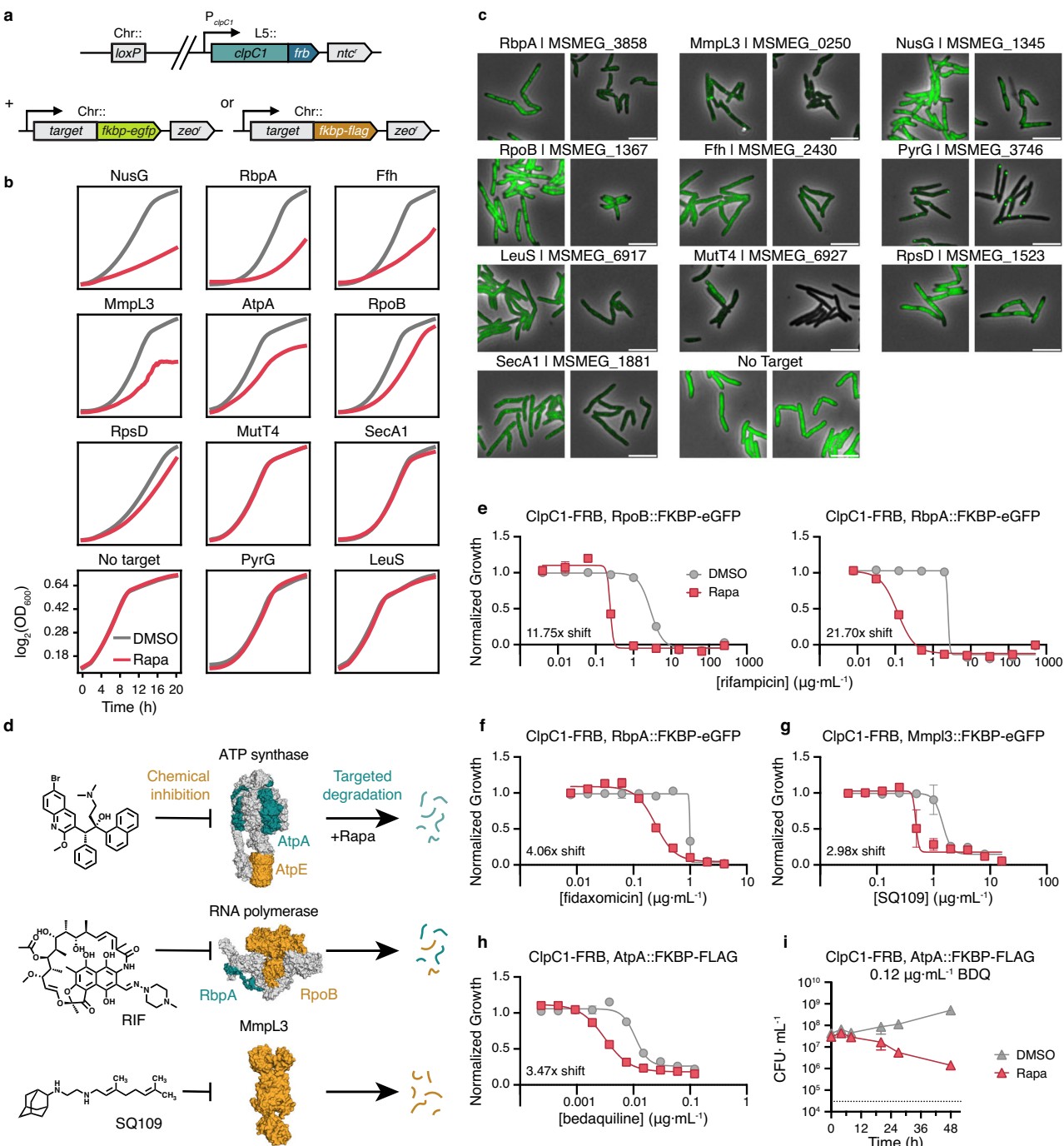

**Fig. 5 | Targeted degradation of native mycobacterial proteins inhibits bacterial growth and potentiates existing antibiotics. a** Schematics of strains generated to assess phenotypes arising from targeted protein degradation, with target alleles tagged with *fkbp-egfp* (*nusG, rpbA, ffh, mmpL3, rpoB, rpsD, mutT4, secA1, pyrG, leuS*) or *fkbp-flag* (*atpA*) at their chromosomal (Chr::) loci in a Δ*clpC1* L5::*clpC1-frb* background. **b** Optical density at 600 nm of bacterial cultures in (**a**) measuring the growth kinetics of when supplemented with DMSO (gray) or 0.1 µg·mL⁻¹ rapamycin (red). **c** Live cell, wide-field fluorescence microscopy images of cells in (**b**) following 9 h of incubation with DMSO or 0.1 µg·mL⁻¹ rapamycin. Scale bar, 5 µm. **d** Schematic of combination approach to potentiate antibiotic efficacy, in which members of the multi-component complexes ATP synthase (top, PDB: 7NJK) or RNA polymerase (middle, PDB: 6C05) or a single protein MmpL3 (bottom, PDB: 6AJG) are inhibited both with small molecule antibiotics (yellow) and targeted for

degradation by rapamycin. **e–h** Half-maximal minimum inhibitory concentration (MIC₅₀) dose response measuring the sensitivity of indicated strains to the indicated antibiotics in media supplemented with either DMSO or 0.1 µg·mL⁻¹ rapamycin (**e–g**) or 0.5 µg·mL⁻¹ rapamycin (**h**). **i** Kill curves measuring the number of individual colony forming units (CFU) following treatment of strains expressing AtpA-FKBP-FLAG with 0.12 µg·mL⁻¹ bedaquiline when supplemented with either DMSO or 0.5 µg·mL⁻¹ rapamycin. *P*-values were determined by unpaired two-tailed *t*-tests with a Holm-Šídák correction for multiple comparisons. *t* = 27 h, *adjusted-*P* = 0.0496; *t* = 48 h, **adjusted-*P* = 0.0325. For (**b**), data are mean of two technical duplicates plotted with sliding window smoothing using a window size of 8 and representative of two independent experiments. For (**e–i**), data are mean ± s.d. of three technical replicates and are representative of two independent experiments. Source data are provided with this paper.

BacPROTAC, whereby ClpC1 is solely responsible for target binding and unfolding, the ClpC1 unfoldase should be physically occupied until the substrate is fully degraded. By contrast, a PafA-targeting BacPRO-TAC would deliver cognate substrates to PafA to be pupylated, after which PafA could, in principle, be recycled more quickly for the next round of substrate labeling. However, the PafA ligase and the bacterial proteasome likely have their own specificities and might only be appropriate for a restricted class of proteins[50]. Given such fundamental differences in these degradation complexes, the designing principles of ClpC1- or PafA-targeting BacPROTAC compounds are foreseeably very different; therefore, a proof-of-principle for the latter should be established to further investigate the PPS's suitability for a TPD approach in mycobacteria.

Using microscopy, we noted that for some, but not all targets, rapamycin addition resulted in the formation of fluorescent puncta. This raises the possibility that delivery of target proteins to ClpC1 could have a dual effect: first, proximity-mediated degradation of that protein, and second, forced removal of that protein from its native localization (Supplementary Fig. 8c), potentially perturbing optimal function or complex formation. This forced removal may be sufficient to cause unfavorable outcomes in bacteria even without substantial degradation; for example, Ffh was only degraded by about 30% of its steady state levels (Supplementary Fig. 7d), yet we observed substantial growth delays in liquid media when degraded (Fig. 5b). There is some evidence for this concept in *Mtb*, where it has been shown that small molecules that disrupt a native protein-protein interaction show strong bactericidal activity[51]. We posit that in addition to TPD, chemically-induced protein re-localization and aggregation may also serve as promising revenues for novel antibiotic discovery; however, further studies are needed to evaluate the feasibility and limitations of these methods, which are all based on the chemical perturbation of endogenous protein-protein interactions.

BacPROTACs exhibit great potential for the modular design of next-generation antimicrobials, as was demonstrated by our study and others previously[11,12]. Nevertheless, technical hurdles in chemical binder screening, chemical synthesis of large modularized compounds, pharmacological optimization for improved membrane permeability and oral availability, host toxicity, and other properties pose major challenges in developing a drug-like BacPROTAC compound[48]. Moreover, there exists only limited information regarding the frequency and mechanisms of BacPROTAC resistance[11,12]. While we envision that the degradation potential of novel essential substrates, as predicted by our machine learning model, may facilitate target prioritization (Fig. 4e, Supplementary Data 3), the druggable potential of these targets would still require orthogonal validation, which could still be resource-demanding. In this case, prioritizing degradation-prone proteins which are also critical components of established drug targets could be advantageous, as BacPROTAC compounds that induce TPD of these substrates might further sensitize the bacteria to existing antibiotics and potentially accelerate the clearance of infection.

## Methods

### Bacterial strains and culture conditions

*Mycobacterium smegmatis* mc²155 and derivative strains were grown in Middlebrook 7H9 liquid media (Millipore) supplemented with ADC (5 g·L$^{-1}$ bovine albumin fraction V, 2 g·L$^{-1}$ dextrose, 0.003 g·L$^{-1}$ catalase, 0.85 g·L$^{-1}$ NaCl), 0.2% (v/v) glycerol, and 0.05% (v/v) Tween-80 or plated on LB agar or 7H10. Antibiotic selection concentrations for *Msm* were: 25 μg·mL$^{-1}$ kanamycin, 20 μg·mL$^{-1}$ nourseothricin, 50 μg·mL$^{-1}$ hygromycin, and 20 μg·mL$^{-1}$ zeocin. Antibiotic selection concentrations for *Escherichia coli* were: 50 μg·mL$^{-1}$ kanamycin, 40 μg·mL$^{-1}$ nourseothricin, 100 μg·mL$^{-1}$ hygromycin, and 50 μg·mL$^{-1}$ zeocin. All strains were cultured at 37 °C. Rapamycin (Sigma) was dissolved in DMSO to a stock concentration of 1 mg·mL$^{-1}$ or 10 mg·mL$^{-1}$. Plasmids and strains generated in this study are summarized in

Supplementary Data 4 and 5. Primers used in this study are summarized in Supplementary Data 6.

### Strain construction

**mc²155 Δ*clpC1* L5::*clpC1-frb* strain.** The *clpC1-frb* strain was made by L5 allele swap. First, a *clpC1* (*msmeg_6091*) merodiploid strain was generated by amplifying *clpC1* with 500 bp of the sequence upstream of the gene to capture the native promoter, assembling into a kan$^r$ plasmid with the L5 integrase, and integrating as a single copy at the L5 phage chromosomal attachment site (*attB*)[52]. Native *clpC1* was knocked out by mycobacterial recombineering (phage-mediated homologous recombination)[53], replacing the native *clpC1* allele with a zeocin selection marker, flanked by *loxP* sites. The knockout construct comprised ~500 bp homology arms upstream and downstream of *clpC1* surrounding the floxed zeocin selection marker and was assembled into a pL5 vector[54]. Next, *clpC1* (with native promoter) and *frb* codon-optimized for mycobacteria were amplified, assembled into a ntc$^r$ plasmid with the L5 integrase, and transformed into the Δ*clpC1* L5::*clpC1* strain; successful swaps acquired nourseothricin resistance at the expense of kanamycin resistance, whereas double integrants contained both markers. A 1:1 ratio of single to double integrants indicates that there is no fitness advantage to either allele.

**Fluorescent *target-fkbp-egfp* fusions.** The genes listed in Fig. 3e, Supplementary Figs. 7, 10, and Supplementary Data 2, as well as *fkbp* and *egfp* codon optimized for mycobacteria were amplified, assembled under control of the high-strength P$_{UV15TetO}$ into a kan$^r$ plasmid with the Tweety integrase, and transformed in the WT *clpC1* or Δ*clpC1* L5::*clpC1-frb* background. All gene IDs for the genes used in this study are listed in Supplementary Data 2.

**Chromosomal locus *target-fkbp* fusions.** The genes listed in Fig. 4e (red filled) and Fig. 5b were tagged with codon-optimized *fkbp-flag* (for AtpA only) or *fkbp-egfp* (the remaining chromosomally tagged strains) at their native locus by recombineering, by adding the coding sequence in frame immediately before each gene's stop codon. First, the zeocin marker at the *clpC1* chromosomal locus was removed by transformation with a Cre plasmid that is sucrose curable by SacB[55]. The knock in construct comprised 500 bp homology arms upstream and downstream of each gene's stop codon surrounding the floxed zeocin marker and was assembled into a pET-26b+ vector. All plasmids used in this work were cloned using isothermal assembly[56] and plasmid maps are available upon request.

**Transformation.** *E. coli* DH5α or XL1b competent cells were prepared using rubidium chloride and transformed by heat shock. Electro-competent *Msm* cells were prepared by growing cells to mid-log phase (OD$_{600}$ ≈ 0.5), washing 3× with 10% (v/v) glycerol pre-chilled to 4 °C, then resuspending in 1/100$^{th}$ of the initial culture volume with 10% glycerol. DNA (200 ng for integrating plasmids, 1 μg for recombineering) was added to 100 μL competent cells and incubated on ice for 5 min, followed by electroporation in a 2 mm cuvette at 2500 V, 125 Ω (plasmids; 1000 Ω recombineering), 25 μF using an ECM 630 electroporator (BTX). 7H9 media (1 mL) was added to the cells, which recovered for 3 h with shaking at 37 °C, before spreading on LB agar and incubating at 37 °C for 2–3 days.

**Rifampicin-resistant strains.** The rifampicin resistant strains (Supplementary Fig. 13b) were generated from the parent *clpC1-frb*, *rbpA::fkbp-egfp* and *clpC1-frb*, *rpoB::fkbp-egfp* strains. For each strain, 1 mL of *Msm* cells at OD$_{600}$ = 1.0 was plated on each of several 7H10 agar plates containing 100 μg·mL$^{-1}$ rifampicin and incubated for four days. A 462 bp region of *rpoB* containing the rifampicin resistance determining region was then amplified from individual colonies and sequenced to confirm the presence of rifampicin-resistance mutations.

## NanoLuc assay

To validate rapamycin-mediated FRB-FKBP dimerization in mycobacteria, *Msm* cells were grown to mid-log phase ($OD_{600} \approx 0.5$) and plated at a final $OD_{600} = 0.1$ with 5-fold serial dilutions of rapamycin. The Nano-Glo® 5× reagent (furimazine substrate + buffer; Promega) was prepared and added to each sample. Luminescence was measured in opaque white plates after incubation with linear shaking at 37 °C for 5 min using a Spark 10 M plate reader (Tecan).

## Protein structure modeling

The crystal structures of the stabilized mutant *Mtb*ClpC1 hexamer (PDB: 8A8U)[21], the *Msm* ATP synthase complex (PDB: 7NJK)[57], the *Mtb* RNA polymerase holoenzyme in complex with RbpA (PDB: 6C05)[42], and *Msm*MmpL3 (PDB: 6AJG)[58] were modeled using PyMOL (Schrodinger). The *Mtb*ClpC1 NTD trimer was modeled by aligning three individual structures of the *Mtb*ClpC1 NTD (PDB: 6PBS)[59] with a *E. coli* ClpB NTD trimer (PDB: 6OG3)[60] as previously in ref. 21.

## Protein expression and purification

The *Mtb* *clpC1* (*rv3596c*) allele and *frb* codon optimized for mycobacteria were amplified and assembled to encode both N- and C-terminally fused gene products under control of the T7 promoter into a pET-26b+ plasmid, which encodes a 6xHis tag at the C-terminal end of the gene product. Constructs were transformed into BL21-CodonPlus(DE3)-RP cells (Agilent), which enables efficient expression of GC-rich genes in *E. coli*.

BL21 strains were grown in LB broth (Lennox) with shaking at 37 °C, until they reached $OD_{600} \approx 0.8$; protein expression was induced overnight with 100 μM IPTG and incubating with shaking at 18 °C overnight. Cells were harvested by centrifugation, subjected to a −80 °C freeze-thaw cycle, and resuspended in protein binding buffer (25 mM Gomori buffer pH 7.6, 100 mM KCl, 5% (v/v) glycerol). Next, cells were treated with 12.5 μg·mL⁻¹ lysozyme for 5 min and then lysed by sonication (30% power pulses for 30 s, alternating with ice incubation for 1 min; repeat 4×). Cell debris was removed by centrifugation and His-tagged ClpC1 proteins were purified from supernatants with Ni-NTA agarose beads (Qiagen). After washing in binding buffer supplemented with 20 mM imidazole, proteins were eluted with binding buffer supplemented with 100 mM and 200 mM imidazole. Input, flow through, and both elutions were examined by SDS-PAGE by loading samples in a NuPAGE 4−12% Bis-Tris precast gradient gel (Invitrogen) using MES SDS buffer.

Eluted proteins were concentrated and desalted with an Amicon® Ultra-4 Centrifugal Filter Device (30 kDa MWCO, Millipore Sigma), with equilibration in protein binding buffer lacking imidazole; protein concentrations were ascertained with the Pierce™ Coomassie Plus (Bradford) Assay Reagent (ThermoScientific) and comparison to a BSA standard curve. Purified fractions were pooled, aliquoted, and stored at −80 °C. FRB-tagged *Mtb*ClpC1 purification was conducted as one independent experiment and WT *Mtb*ClpC1, *Mtb*ClpP1, and *Mtb*ClpP2 were purified previously[22,61].

## In vitro assays

**ClpC1P1P2 protease activity.** Degradation of eGFP-ssrA by purified *Mtb*ClpC1P1P2 was measured by examining loss in fluorescence signal over time. *Mtb*ClpC1 (WT and FRB-tagged proteins, 0.26 μM hexamer), *Mtb*ClpP1P2 (0.26 μM tetradecamer), and eGFP-ssrA (1.25 μM) were mixed in buffer containing: 25 mM Gomori buffer pH 7.6 supplemented with 100 mM KCl (Sigma), 5% glycerol (Sigma), 8 mM Mg-ATP (Sigma), and 2.5 mM Bz-Leu-Leu[62] (Laboratory of Dr. William Bachovchin, Tufts University). Fluorescence signal decay was measured (Ex/Em: 485 nm / 520 nm) continuously in a SpectraMax M5 microplate reader (Molecular Devices) at 37 °C for 35 min.

**ClpC1 ATPase activity.** ATP hydrolysis by purified *Mtb*ClpC1 was measured using a pyruvate kinase and lactate dehydrogenase (PK/ LDH) coupled enzymatic assay in which absorbance at 340 nm is an inverse proxy for the rate of ATP hydrolysis. *Mtb*ClpC1 (WT and FRB-tagged proteins, 0−0.7 μM hexamer), 1 mM phosphoenolpyruvate (Sigma), and 20 U·mL⁻¹ PK/LDH (Sigma) were mixed in buffer containing: 50 mM Tris pH 7.6 (Sigma), 100 mM KCl (Sigma), 1 mM DTT (Sigma), 1 mM NADH (Sigma), and 4 mM Mg-ATP (Sigma). $Absorbance_{340}$ was measured continuously in a SpectraMax M5 microplate reader (Molecular Devices) at 37 °C for 32 min.

## Flow cytometry

To measure the fluorescence of the target reporter fusions in the presence of rapamycin, *Msm* cells were grown to mid-log phase ($OD_{600} \approx 0.5$) and plated at a final $OD_{600} = 0.2$ (for the time course flow cytometry, $OD_{600} = 0.1$) with the denoted concentration of rapamycin or DMSO (vehicle control). The flow cytometry screening was carried out in technical duplicate where the same bacterial culture was inoculated into replicate wells containing either DMSO or rapamycin. Cells were incubated with shaking at 700 rpm at 37 °C for the specified times (0 h, 3 h, 6 h, 9 h, 12 h), diluted in 7H9 media (to remain under 10,000 events·sec⁻¹), and fluorescence was quantified on a MACSQuant® VYB flow cytometer using the green fluorescence laser and filter set (channel B1, Ex/Em: 488 nm / 525/50 nm) with at least 10,000 events captured for each sample. To suppress spurious events due to noise or cell debris, event triggers were applied on forward scatter peak height (FSC-H > 0.8) and side scatter area (SSC-A > 1.2). To exclude cellular aggregates and morphological outliers, gates were applied with FlowJo (v. 10.8), and a representative gating strategy can be found in Supplementary Fig. 6a. The remaining events were normalized to mode and presented as histograms displaying $log_{10}$-transformed green fluorescence intensity area (AlexaFluor488-A), using GFP signal as a proxy for target protein levels in vivo. Full flow cytometry plots for all the training and validation reporter strains are included in Supplementary Figs. 7c and 10c, respectively.

## Western blots

To directly examine protein levels when redirected to ClpC1-FRB, *Msm* cells were grown to mid-log phase ($OD_{600} \approx 0.5$), pelleted by centrifugation, and resuspended in 1× TBS supplemented with the cOmplete™ Protease Inhibitor Cocktail (Roche). Resuspended cells were aliquoted into FastPrep® 2 mL Lysing Matrix tubes (MPBio) and lysed by beat beating with a Precellys 24 tissue homogenizer (Bertin Technologies; 6,500 rpm for 45 s, alternating with ice incubation for 2 min; repeat 3×). Cell debris was removed by centrifugation, supernatant protein content was evaluated by NanoDrop 1000 (ThermoScientific), and all samples were normalized to the same protein concentration in 1× TBS with protease inhibitor. Normalized protein aliquots were treated with TURBO™ DNase (Invitrogen) for 15 min at 37 °C to digest mycobacterial genomic DNA and then denatured with boiling in Laemmli buffer. For all samples, 140 μg of protein and PageRuler™ Prestained Protein Ladder (ThermoScientific) was loaded onto NuPAGE 4−12% Bis-Tris precast gradient gels (Invitrogen) using MES SDS buffer, transferred to PVDF membranes using a Trans-Blot Turbo Transfer System (Bio-Rad), and blocked with 1:1 SEA-BLOCK blocking buffer (ThermoScientific) and 1× PBS with shaking for 1 h at 25 °C. Membranes were probed with the specified antibodies in 1:1 SEA-BLOCK and 1× PBS-Tween (phosphate-buffered saline with 0.1% (v/v) Tween-20) with shaking for 1 h each primary and secondary at 25 °C. Western blots were imaged using the Odyssey® CLx Infrared Imaging System (LI-COR) and processed using LI-COR ImageStudio. The full uncropped scans of the blot shown in Fig. 2e can be found in Supplementary Fig. 5. The mouse IgG2b monoclonal RpoB antibody (Invitrogen, clone #8RB13) was diluted 1:1,000[63]. The rabbit IgG monoclonal GFP antibody (Abcam, clone #EPR14104) was diluted 1:1,000[64]. The IRDye® 680RD goat IgG (H + L) anti-mouse and IRDye®

800CW goat IgG (H + L) anti-rabbit (LI-COR) fluorescent secondary antibodies were diluted 1:15,000.

## Time-lapse fluorescence microscopy and data analysis

To generate the time-lapse data for targeted degradation of RpoA-FKBP-eGFP in Fig. 2f, cells were grown until late log phase ($OD_{600} \approx 1.0$)) and seeded on a 2.0% agarose pad containing 1× concentration of 7H9, 0.1% w/v casamino acid (BD), 0.2% w/v glucose, 0.2% v/v glycerol and 0.1 μg·mL$^{-1}$ rapamycin. The agarose pad was cast in a $12 \times 12$ mm$^2$ customized plastic frame and placed in a low-evaporation imaging disk (MatTek Corp.) to enable long-term time-lapse experiments. Fluorescence and phase-contrast images were acquired every 10 min over a 9 h period with a 100× Plan Apo oil objective (NA = 1.45) using a Nikon Ti-E inverted, widefield microscope equipped with a Nikon Perfect Focus system with a Piezo Z drive motor, Andor Zyla sCMOS camera, and an Agilent MLC400 Monolithic laser combiner. Fluorescence signals were acquired using a 6-channel Spectra X LED light source and the Sedat Quad filter set. The excitation and emission filters used for the green fluorophores (eGFP) were Ex. 470/24 nm and Em. 515/25 nm. Time-lapse snapshots were rendered using customized Python scripts based on the Python package *microfilm* (https://github.com/guiwitz/microfilm). To analyze time-lapse data, our previously established microscopy image analysis pipeline MOMIA[65] was modified to track the fluorescence of individual cells and emergence of new cell generations. Briefly, full-sized (2048 × 2048) microscopy image stacks were manually cropped to capture the growth of individual micro-colonies. A Residual-Attention-Unet model[66] was trained to perform cell segmentation from phase contrast images. Segmented objects of consecutive frames were linked based on their overlaps in adjacent frames and their resemblance in cell morphology. The initial linkage was refined iteratively to minimize linkage drift. Cellular fluorescence per frame was estimated by the mean intensity of cellular ROI of the fluorescence image, which had been cropped and removed of background fluorescence using the rolling-ball method.

## Target protein microscopy image acquisition

To capture the microscopy data for targeted degradation of all 72 target proteins examined in Fig. 3f and Supplementary Figs. 8c and 10e, cells were sampled from the flow cytometry experiment at 9 h after inoculation and seeded on 2.0% agarose pads. Fluorescence and phase-contrast images were acquired using the same microscope configuration as used for the time-lapse experiments. Image segmentation and fluorescence intensity quantification were performed using a Python analysis pipeline similar to the one used for time-lapse image analysis. Image crops were rendered using customized Python scripts powered by *microfilm* (guiwitz.github.io/microfilm).

## Protein feature extraction

The full list of conserved orthologs shared by *Msm* and *Mtb* was generated by combining ortholog annotations from the Mycobacteria Systems Resources[57] and Mycobrowser[29], which yielded a total number of 2450 protein candidates. Gene essentiality and vulnerability annotations were referenced from the seminal work from Bosch and others[10,28]. After removing non-essential candidates, we obtained 354 essential orthologs. We also included 3 genes (*msmeg_3858/rbpA*, *msmeg_1435/rpsJ*, *msmeg_2299/nrdE*) for which their *Mtb* counterparts were annotated as essential whereas the annotation for these *Msm* copies were ambiguous. Proteins that are shorter than 100 amino acids or longer than 1500 amino acids were manually excluded from the candidate list. For the remaining 348 proteins, AlphaFold prediction confidence scores (pLDDt) were retrieved from the AlphaFold Protein Structure Database (https://alphafold.ebi.ac.uk/)[34,58], and Disorder Potential predictions were downloaded from the flDPnn web server (http://biomine.cs.vcu.edu/servers/flDPnn/)[35]. For each protein, we generated five test subjects: the N- terminal 15 or 30 amino acids, the C-terminal 15 or 30 amino acids, and the full-length protein. For each subject, we generated a comprehensive set of quantitative structure-activity relationship (QSAR) metrics using the Python package *peptides* (https://github.com/althonos/peptides.py) and computed the average and standard deviation of the sectional QSAR, pLDDt, and flDPnn scores. To estimate the length of the unstructured stretch of the termini, we defined the lower or higher bound for flDPnn or pLDDt scores as 0.2 and 50, respectively, and computed the fraction of disordered or poorly predicted terminal residues. The 348 conserved proteins and their corresponding protein features are listed in Supplementary Data 1.

## Machine learning and prediction

Z-score data normalization was performed on each of the 485 numeric features across the 348 proteins. Feature selection was carried out using the SelectKBest function scored of the Python package *Scikit-learn*[67]. Here we used data of the substrates of the first batch (54 proteins in total) as the full training set, which was further split into two subsets at a ratio of 0.67 and 0.33, respectively. The random train-test splitting and SelectKBest scoring were repeated 10 times to find features which better correlate with the log$_2$-transformed degradation constants. The scores were averaged and ranked, and the top 40 features were selected for model training. Multiple Lasso regression models were trained with varying L1 regularization terms on the full training set and evaluated on the validation dataset (comprised of the 18 proteins of the second experimental batch) by Pearson's correlation. The best model achieved a Pearson's correlation coefficient over 0.6 and a statistically significant *P* value (0.0057), The selected model was used to predict the log$_2$-transformed degradation constant as depicted in Fig. 4e.

## *atpA* locus PCR validation

To test for the presence of an *atpA* locus duplication (Supplementary Figs. 12a, b), primers were designed to amplify regions corresponding to the wild-type or tagged locus. A primer pair to amplify a region of *rpoB* was included as a control. For each reaction, 16 ng of genomic DNA from the indicated strains was used as a template and amplified using KOD Xtreme™ Hot Start DNA Polymerase (Sigma-Aldrich).

## Minimum inhibitory concentration (MIC) assay

To calculate their dose response to antibiotics and targeted degradation, *Msm* cells were grown to mid-log phase ($OD_{600} \approx 0.5$) and plated at a final $OD_{600} = 0.0015$ in each well of a 96-well plate containing serial dilutions of the specified antibiotics. For assays with concurrent targeted degradation, all wells were supplemented with the specified concentration of rapamycin or DMSO. Plates were incubated with drug with shaking at 150 rpm at 37 °C overnight. Resazurin was added to a final concentration of 0.002% (w/v), plates were incubated up to 18 h, and fluorescence conversion was measured (Ex/Em: 560 nm/ 590 nm) in a SpectraMax M2 microplate reader (Molecular Devices) or a Spark 10 M plate reader (Tecan). Background fluorescence was subtracted from wells without cells and fluorescence signal was normalized to wells without any compound. A nonlinear regression was used to fit a sigmoid curve to the dose response data and calculate the half-maximal minimum inhibitory concentration (MIC$_{50}$) using GraphPad Prism.

## Antibiotic kill kinetics curves

To examine the effect of targeted degradation on bedaquiline killing, *Msm* cells were grown to mid-log phase ($OD_{600} \approx 0.5$) and plated at a final $OD_{600} = 0.08$ in 24 well plates; wells were supplemented with 0× or 10× their measured MIC of bedaquiline and with 0.5 μg·mL$^{-1}$ rapamycin or DMSO. Plates were incubated for 2 days with shaking at 155 rpm at 37 °C and cells were sampled at 0, 4, 8, 20, 27, and 48 h after plating. Sampled cells were serially diluted and plated on plain LB agar;

plates were incubated for 2–3 days at 37 °C and individual colony forming units (CFUs) were counted.

## Bacterial growth kinetics curves

To examine the effect of targeted degradation on bacterial growth in liquid media, *Msm* cells were grown to mid-log phase ($OD_{600} \approx 0.5$) and plated at a final $OD_{600} = 0.0015$. Wells were supplemented with the specified concentration of rapamycin or DMSO. Absorbance ($OD_{600}$) and/or eGFP fluorescence (Ex/Em: 488 nm/510 nm) was measured in clear 96 well plates sealed with a Breathe-Easy® membrane (Sigma) with linear shaking at 37 °C for 48 h using a Spark 10 M plate reader (Tecan).

## Statistical analysis

The statistical analyses and correlation plots described in this study were performed using the python package SciPy[68] or GraphPad Prism. All experiments were performed at least twice (biological replicates), unless otherwise noted. Technical replicates were measured within the same experiment from distinct samples, unless otherwise noted. Means were compared using a two-tailed Student's *t*-test with a Holm-Šídák correction. Adjusted $P < 0.05$ was considered significant. For the microscopy experiments at least 70 cells were analyzed for each shown image and the image data shown are representative of multiple fields. No statistical methods were used to determine sample sizes used in this work and the investigators were not blinded to sample identity.

## Software

Fiji (1.52p), Scikit-learn (1.0.1), Scikit-image (0.19.3), OpenCV2 (4.5), NumPy (1.22.4), Tensorflow (2.6), Keras (2.6), Pandas (1.3.4), and Python (3.8) to analyze plate and microscopy images; SciPy (1.10.0), Matplotlib (3.6.2) and GraphPad Prism (10.0.3) were used to perform statistical analysis and to create data graphs. Flowjo (10.8.2), NumPy, and SciPy were used to analyze flow cytometry data. Peptides.py (v0.3.2) was used to extract sequence features from proteins and oligopeptides.

## Data availability

Example unprocessed microscopy images, clips of the timelapse microscopy data, and flow cytometry data generated in this study have been deposited in the Zenodo database under the following https://doi.org/10.5281/zenodo.11004395. Due to size limitations, the rest of the microscopy data are available upon request. AlphaFold protein structure files were downloaded from the AlphaFold Protein Structure Database and are available at https://alphafold.ebi.ac.uk. Source data are provided with this paper.

## Code availability

All code used to generate primers and analyze data has been deposited on Zenodo and is available under the https://doi.org/10.5281/zenodo.11004395.

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

## Acknowledgements

We thank members of the Eric J. Rubin and Sarah Fortune labs for helpful discussions during this work and Douaa Mugahid for thoughtful feedback on the manuscript. We thank Kenan Murphy for sharing the ClpC1-eGFP *Msm* strain. Funding: This material is based upon work supported by the NSF Graduate Research Fellowship under Grant No. DGE1745303 (H.W.), the Harvard GSAS Herchel Smith Graduate Fel-lowship (H.W.), the Marcus Urann Graduate Fellowship (H.W.), the National Institute of General Medical Sciences award T32GM007753 and T32GM144273 (S.Z.), the NIH T32 Molecular, Cellular, and

Developmental Mechanisms grant 5T32GM145407 (S.Z.), the NSF Graduate Research Fellowship under Grant No. DGE2140743 (E.S.), the NIH/NIAID award U19 AI142735 (E.J.R.), and funding provided by Grace Wang and Josef Tatelbaum. The content is solely the responsibility of the authors and does not necessarily represent the official views of the National Institute of General Medical Sciences or the National Institutes of Health.

## Author contributions
H.W., S.Z., J.Z., and E.R. conceived and designed the study; H.W., J.Z., S.Z., O.K., J.S., and MW performed experiments; H.W., J.Z., S.Z., E.S., and I.W. performed data analysis; O.K., T.A., M.C., E.R. gave technical and conceptual advice; H.W. and J.Z. wrote the original draft; H.W., J.Z., S.Z., M.C., I.W., O.K., T.A., J.S., M.W., E.S., and E.R. reviewed and edited the manuscript; J.Z. and E.R. supervised this study.

## Competing interests
The authors declare no competing interests.
