## [Peer Review File · Nature Communications]

Reviewers' Comments:

Reviewer #1:

Remarks to the Author:

Targeted protein degradation with the use of PROTACs (proteolysis targeting chimeras) has emerged as a new therapeutic avenue, in particular for drug targets considered undruggable by conventional methods. More recently it has been demonstrated that, in principle, PROTACs could be used as a new class of antibiotics, targeting essential bacterial proteins for degradation. *Mycobacterium tuberculosis* (Mtb), the pathogen that causes tuberculosis and kills more than a million people each year, presents a particularly important target, since multi-drug resistant Mtb strains are an increasing problem worldwide.

The manuscript by Won et al is interesting, because it addresses the crucial question of how susceptible essential proteins in *Mycobacteria* are to the PROTAC approach. This aspect of the PROTAC approach in bacteria has received less attention as the field is still focused on establishing the principle of bacterial PROTACS. The authors use the rapamycin-inducible heterodimerization between FRB and FKBP, two small proteins that both bind to different regions of the macrolide rapamycin. An *M. smegmatis* strain was produced where ClpC1 was replaced with a ClpC1-FRB fusion. A rational selection of 16 proteins known to be essential for Mtb growth was then introduced from an integrative plasmid as an N-terminal FKBP-eGFP fusion for assessment of degradation rate by high-throughput flow cytometry screening. Inhibition of a subset of these was investigated by replacing the genomic copies with FKBP-eGFP fusions in the ClpC1-FRB strain. The idea and concept of the study presents an approach to the topic of bacterial PROTACs from the point of view of the TDP targets that is very timely and complements existing approaches trying to establish the bifunctional chemistry for bacPROTACs.

However, the choice and scope of substrates chosen for the study limit its predictive usefulness and preclude the reader from drawing generalized conclusions. It would be helpful to broaden the screen in order to allow for more meaningful conclusions to be drawn from this approach.

Comments:

1. The aim of the study is to use a proximity screen in order to identify endogenous Mtb substrates that are amenable to TPD. As such one would expect that a significant amount of the known essential proteome of Mtb is screened. However, in the end the ClpC1-FRB strain is transformed with plasmids for expression of only 16 different endogenous protein targets that are fused to the FKBP-eGFP for proximity induced protein degradation, three of which belong to the same protein complex, namely ATP synthase. In order to obtain a statistically more relevant sample that allows more generalized conclusions, it would be useful to screen a larger number of potential targets (at least three times the current size, better more).

2. There should be more overlap between the targets looked at by the proximity TPD screening and the targets assessed for bacterial growth inhibition. Only six of the 16 chosen potential targets are tested in terms of effect on bacterial growth that can be achieved when subjecting them to TPD. It is also not clear why the selection was chosen in this way, since none of the really good degradation targets are tested (e.g. HupB, RbpA, AccD5). Not surprisingly, only one of the proteins causes a significant growth defect when subjected to TPD. The authors should at the very least include the three most promising degradation substrates (as judged by their proximity-induced degradation screen).

3. In order to increase the number of targets for which they observe a growth defect, the authors include AtpA, another subunit of ATP synthase. However, exactly this subunit was not used in the proximity assessment screen. The authors should definitely test AtpA in the flow cytometry screen.

4. The correlation with the currently used sets of targets in the two assays is rather poor. This would speak against the screening strategy as a predictive tool for good targets to invest more effort into. Again, the low numbers in each data set preclude a meaningful analysis, what the reasons for this lack of correlation.

Minor comments:

- For those targets where fluorescence decreases only mildly or not at all, is it due to failure to degrade the target or is it because GFP cannot be pulled in with it?
- Figure 2 c,d: please add axis labels/units
- Line 298-302: I do not understand this sentence and the point the authors are trying to make here. Please explain!
- Line 301: which histidine do the authors refer to here?

Reviewer #2:

Remarks to the Author:

This creative manuscript describes a chemical genetic approach to examine the feasibility of targeted protein degradation in mycobacteria. Studying induced proximity to ClpC1P1P2 proteolytic system through the creation of ClpC1-FRB and target-FKBP fusions, which dimerize when rapamycin is added to the *M. smegmatis* cells. The approach is shown to be robust and the susceptibility to targeted protein degradation of 17 mycobacterial targets is examined.

Key results, in addition to the elegant development of the FKBP – ClpC1 reporter system, include selective degradation of exogenous Msm proteins with a propensity different from the natural substrates of the mycobacterial Clp system, the system allows for the identification and prioritization of targets most susceptible to the TPD approach to be prioritized for PROTAC development, potentiation of antibiotics by targeting proteins with close links to the antibiotic target, and induced cellular co-location in puncta.

Overall, the manuscript provides an important step forward in developing targeted protein degradation strategies for mycobacteria and is complementary to chemical approaches recently published.

The rigor of the reported studies and associated experimental data is high, and there are no concerns about the reproducibility of the results.

Minor suggested revisions and comments for the authors to consider follow.

1. Line 211. Orthogonal proof of concept and selectivity of target degradation in TPD systems is best demonstrated by unbiased proteomic experiments such as using TMT method. The addition of such an experiment here would strengthen this manuscript. As presented, it is possible, though unlikely, that the antimicrobial effect is due, in part, to the degradation of other proteins.
2. Line 219. Resistance development and host toxicity are the big unknowns PROTAC in Eukaryotic systems as this new modality progress in the clinic. Given the high propensity for bacteria to gain resistance and restrictions to drug uptake, developing bacterial PROTACs may not be as easy as stated.
3. Line 322. Is there any correlation between the folding tightness (DSF T_m) and the ability to be degraded.
4. Did the authors consider degrading drug-activating enzymes to demonstrate antagonism such as EthA since InhA is not susceptible? or transcription factors. It would be good to know the rationale for the choice of the 17 targets studied in this manuscript.

Reviewer #3:

Remarks to the Author:

The manuscript of Won and colleagues describes the application of targeted protein degradation (TPD) in *Mycobacterium smegmatis* in an approach to examine bacterial protein proteolysis as a new therapeutic for tuberculosis. This manuscript is timely as new interventions for treating drug resistance in tuberculosis are urgently required. It follows the application of recently developed Bacterial PROTACs but utilising a chemically induced system using the small molecule rapamycin instead of specific chemical ligands used in PROTAC strategies. The work is novel and sheds light

into the application of TPD in mycobacteria. The authors analysed using the system 17 essential genes and interestingly they found differences in their suitability of being targets for TPD. Overall, it is a well planned and robust study with careful experimentation and appropriate controls were used throughout. In addition to identifying targets that are suitable for PROTAC based degradation, the authors also identified that this strategy could potentiate the outcomes of current antibiotics with proteins of the same pathway. This conclusion is based on the potentiation of bedaquiline and rifampicin when AtpA and RpoA were targeted consequently. Even though this might be true for these two cases, I think the authors should have tested more drugs/gene combinations to conclude and strengthen this hypothesis further.

Minor comments:

A table with the selected genes and the rationale behind the selection would be a good addition.

Typo in line 338: we were able...

More details in the method (lines 569-571) required so that it is clear how these images were captured and their subsequent analysis.

RESPONSE TO REVIEWERS' COMMENTS

Manuscript ID: NCOMMS-23-06793

Manuscript title: *Targeted Protein Degradation in Mycobacteria Uncovers Antibacterial Effects and Potentiates Antibiotic Efficacy*

Dear editor and reviewers:

We would like to thank you for your thorough and insightful comments on our submitted manuscript. We have revised our manuscript according to the reviewers' comments and suggestions and have provided additional data to address the questions and concerns kindly raised by the reviewers. Here is a summary of some notable changes to the manuscript:

1. we expanded our reporter library to cover 72 conserved substrates with diverse functional and biochemical properties, which enabled us to train a machine learning model that could predict a protein's degradation potential from its sequence and structural features. These new results are now illustrated in Figs. 3-4 and their corresponding supplementary figures.
2. We generated more *M. smegmatis* strains to evaluate the physiological consequences of targeted degradation of endogenous proteins. We replaced colony growth dynamics with liquid culture growth data for improved interpretability. These results are now depicted in Fig. 5.
3. During the revision process, we were notified by a colleague that our lab strains of *M. smegmatis* might contain a gene duplication in its ATP synthase locus. We examined our strain collection with chromosomally tagged ATP synthase component (AtpA, AtpG, etc.) by PCR and nanopore sequencing and found that the FKBP-eGFP tagged strains were indeed merodiploid. By contrast, the AtpA-FKBP-Flag strain, which was used to assess bedaquiline synergy, had only one copy and was faithfully tagged with FKBP-Flag (Supplementary Fig. 13a, b). For this reason, we removed original figure panels that were based on the merodiploid variants and retained only data generated using the AtpA-FKBP-Flag strain (Fig. 5b, h, i).
4. Because these extensive changes required many additional experiments, we have recruited additional help and altered the list of authors.

These changes will be elaborated below with colored text, and we must express our appreciation again for all the fantastic questions and suggestions, which we believe have greatly improved this work.

Sincerely,

Authors of manuscript **NCOMMS-23-06793**.

Reviewer #1 (Remarks to the Author):

Targeted protein degradation with the use of PROTACs (proteolysis targeting chimeras) has emerged as a new therapeutic avenue, in particular for drug targets considered undruggable by conventional methods. More recently it has been demonstrated that, in principle, PROTACs could be used as a new class of antibiotics, targeting essential bacterial proteins for degradation. Mycobacterium tuberculosis (Mtb), the pathogen that causes tuberculosis and kills more than a million people each year, presents a particularly important target, since multi-drug resistant Mtb strains are an increasing problem worldwide.

The manuscript by Won et al is interesting, because it addresses the crucial question of how susceptible essential proteins in Mycobacteria are to the PROTAC approach. This aspect of the PROTAC approach in bacteria has received less attention as the field is still focused on establishing the principle of bacterial PROTACS. The authors use the rapamycin-inducible heterodimerization between FRB and FKBP, two small proteins that both bind to different regions of the macrolide rapamycin. An *M. smegmatis* strain was produced where ClpC1 was replaced with a ClpC1-FRB fusion. A rational selection of 16 proteins known to be essential for Mtb growth was then introduced from an integrative plasmid as an N-terminal FKBP-eGFP fusion for assessment of degradation rate by high-throughput flow cytometry screening. Inhibition of a subset of these was investigated by replacing the genomic copies with FKBP-eGFP fusions in the ClpC1-FRB strain.

The idea and concept of the study presents an approach to the topic of bacterial PROTACs from the point of view of the TDP targets that is very timely and complements existing approaches trying to establish the bifunctional chemistry for bacPROTACs. However, the choice and scope of substrates chosen for the study limit its predictive usefulness and preclude the reader from drawing generalized conclusions. It would be helpful to broaden the screen in order to allow for more meaningful conclusions to be drawn from this approach.

Comments:

1. The aim of the study is to use a proximity screen in order to identify endogenous Mtb substrates that are amenable to TPD. As such one would expect that a significant amount of the known essential proteome of Mtb is screened. However, in the end the ClpC1-FRB strain is transformed with plasmids for expression of only 16 different endogenous protein targets that are fused to the FKBP-eGFP for proximity induced protein degradation, three of which belong to the same protein complex, namely ATP synthase. In order to obtain a statistically more relevant

sample that allows more generalized conclusions, it would be useful to screen a larger number of potential targets (at least three times the current size, better more).

Reply: We greatly appreciate your insightful comment. Indeed, we shared the same concern that the number of substrates evaluated in our first submission was insufficient to guide future study. To address this limitation, we compiled a list of 348 conserved mycobacterial protein which are 1. Highly conserved between *Msm* and *Mtb* and 2. Required for normal growth in both organisms according to established gene essentiality (reference #10 of the present manuscript) and/or transcriptional vulnerability (reference #29) datasets. These proteins, including the 16 substrates tested in our first submission, are now listed in Supplementary Table 1. From the shortlist of 348 conserved essential proteins, we semi-randomly selected 58 additional proteins with diverse functional and structural properties (Supplementary Figs. 7a-b, 10a-b and Supplementary Table 2) and generated reporter strains correspondingly. We took Wag31 and HupB off the reporter strain list as they had to be expressed from a weak promoter to mitigate toxicity, which might introduce unnecessary complications to downstream analysis. The full candidate list now contains 72 proteins (including the previous 14). To minimize batch variation, we repeated the flow-cytometry screening assay for all the 72 substrates. Data pertaining to the expanded reporter library are summarized in Fig. 3, Fig. 4, and Supplementary Figs. 7, 8, and 10.

2. There should be more overlap between the targets looked at by the proximity TPD screening and the targets assessed for bacterial growth inhibition. Only six of the 16 chosen potential targets are tested in terms of effect on bacterial growth that can be achieved when subjecting them to TPD. It is also not clear why the selection was chosen in this way, since none of the really good degradation targets are tested (e.g. HupB, RbpA, AccD5). Not surprisingly, only one of the proteins causes a significant growth defect when subjected to TPD. The authors should at the very least include the three most promising degradation substrates (as judged by their proximity-induced degradation screen).

Reply: We agree with the reviewer that in our first submission, the targets we selected for assessing bacterial growth inhibition were not fully supported by our reporter data. During the revision process, we strived to chromosomally tag more and functionally diverse proteins with the FKBP moiety. As homologous recombineering is substantially less efficient than phage

integrase mediated genome insertion (which was used to build the reporter library), we only obtained 8 out of over 20 candidate substrates. We did, however, successfully tag RbpA, which was the best substrate by reporter assay, and NusG and MutT4, which were predicted to be efficiently degraded by our system. We found that the degradation of NusG and RbpA strongly inhibited bacterial growth, and that the degradation of RbpA indeed sensitized mycobacteria to rifampicin by nearly 20-fold. These data are now re-organized and illustrated in Fig. 5.

3. In order to increase the number of targets for which they observe a growth defect, the authors include AtpA, another subunit of ATP synthase. However, exactly this subunit was not used in the proximity assessment screen. The authors should definitely test AtpA in the flow cytometry screen.

Reply: Thank you for pointing this out. We have now included AtpA in our reporter strain library (Fig. 3d, e and Supplementary Table 2) and have confirmed that it to be moderately susceptible to proximity induced degradation by ClpC1.

4. The correlation with the currently used sets of targets in the two assays is rather poor. This would speak against the screening strategy as a predictive tool for good targets to invest more effort into. Again, the low numbers in each data set preclude a meaningful analysis, what the reasons for this lack of correlation.

Reply: Thank you for the insightful comment. We have now included a plot of the correlation between measured degradation via fluorescence and predicted degradation constant for the chromosomally tagged strains (Supplementary Fig. 12e). We hope this allows us to leverage the measured degradation data from the expanded set of 72 tested substrates as well as useful protein features to better predict suitable targets for future work.

Minor comments:

- For those targets where fluorescence decreases only mildly or not at all, is it due to failure to degrade the target or is it because GFP cannot be pulled in with it?

Reply: Thank you for this great question. Based on our Lasso regression analysis (Fig. 5a-c, Supplementary Fig. 11), nearly all of the poorly degraded substrates contained a “rigid” N terminus, or low disordered propensity (reference #35). We posit that the lack of a floppy N terminus may restrict their accessibility to the ClpC1 unfoldase and render them resistant to proximity mediated degradation. Our imaging data depicted in Supplementary Fig. 8c also

suggest that for a subset of these degradation-resistant proteins (for instance, MSMEG_3144, MSMEG_6933), their cellular fluorescence pattern upon rapamycin treatment appeared very different from that of free FKBP-GFP, implying that these proteins were probably not fragmented upon rapamycin treatment.

- Figure 2 c,d: please add axis labels/units

Reply: Thank you for catching this. We have now added the flow-cytometry fluorescence readings (x-axis) and axis labels accordingly.

- Line 298-302: I do not understand this sentence and the point the authors are trying to make here. Please explain!

- Line 301: which histidine do the authors refer to here?

Reply: We are extremely grateful that you pointed these issues out. Here we will address both questions together: for Lines 298-302 in the original manuscript, we meant to discuss the kinetic differences between ClpC1 and PafA substrate recognition, whereby ClpC1 is occupied throughout the entire degradation cycle whereas PafA may exert its function via a “touch-and-go” mechanism. As for line 301, it should be lysine instead of histidine, and we intended to express that PafA only needs to transfer the Pup peptide onto one lysine residue on the surface of a cognate substrate before releasing the substrate for proteasome degradation. To avoid confusion, we have re-written the paragraph (Lines 360-375).

Reviewer #2 (Remarks to the Author):

This creative manuscript describes a chemical genetic approach to examine the feasibility of targeted protein degradation in mycobacteria. Studying induced proximity to ClpC1P1P2 proteolytic system through the creation of ClpC1-FRB and target-FKBP fusions, which dimerize when rapamycin is added to the *M. smegmatis* cells. The approach is shown to be robust and the susceptibility to targeted protein degradation of 17 mycobacterial targets is examined.

Key results, in addition to the elegant development of the FKB – ClpPC1 reporter system, include selective degradation of exogenous Msm proteins with a propensity different from the natural substrates of the mycobacterial Clp system, the system allows for the identification and prioritization of targets most susceptible to the TPD approach to be prioritized for PROTAC

development, potentiation of antibiotics by targeting proteins with close links to the antibiotic target, and induced cellular co-location in puncta.

Overall, the manuscript provides an important step forward in developing targeted protein degradation strategies for mycobacteria and is complementary to chemical approaches recently published.

The rigor of the reported studies and associated experimental data is high, and there are no concerns about the reproducibility of the results.

Minor suggested revisions and comments for the authors to consider follow.

1. Line 211. Orthogonal proof of concept and selectivity of target degradation in TPD systems is best demonstrated by unbiased proteomic experiments such as using TMT method. The addition of such an experiment here would strengthen this manuscript. As presented, it is possible, though unlikely, that the antimicrobial effect is due, in part, to the degradation of other proteins.

Reply: Thank you for your very kind comments. We agree that quantitative proteomics would be very informative in terms of substrate selectivity, and we consider it as a critical part of our ongoing research to develop BacPROTAC compounds targeting some of the novel candidates uncovered in this work. As the goal of this study was to establish the bulk principles for BacPROTAC target prioritization, we sought to focus more on quantifying the degradation potential of mycobacterial proteins and the phenotypic outcomes if they were to be degraded by ClpC1. Instead of quantitative proteomics, which we thought to be beyond the scope of this work, we provided two orthogonal pieces of evidence to demonstrate that for at least a subset of these proteins, like MmpL3, RpoB, RbpA, and AtpA, their degradation 1. sensitized Msm to antibiotics that target the same pathway but not others and 2. led to changes in cell morphology that phenocopied corresponding gene hypomorphs. These data indicated that our TPD system likely achieved antimicrobial effect by selectively degrading the targeted substrates, or at most, them and their partner proteins belonging to the same functional pathway.

2. Line 219. Resistance development and host toxicity are the big unknowns PROTAC in Eukaryotic systems as this new modality progress in the clinic. Given the high propensity for bacteria to gain resistance and restrictions to drug uptake, developing bacterial PROTACs may not be as easy as stated.

Reply: We greatly appreciate these insightful comments. While we are actively working on developing BacPROTAC compounds that target some of the promising substrates (e.g. RbpA, NusG), we haven't made enough progress to practically test resistance and host toxicity. Nevertheless, we fully agree that these concerns and limitations should be thoroughly evaluated throughout the development pipeline and have added a brief section (Lines 390-403) in the discussion panel accordingly.

3. Line 322. Is there any correlation between the folding tightness (DSF T_m) and the ability to be degraded.

Reply: This is truly a fantastic question! In our first submission, we were limited by the number of tested substrates to conduct a meaningful SAR analysis. With the expanded reporter strain library (from 16 to 72), we compiled a series of protein sequence and structural features to test whether these features possess any power in predicting a protein's degradation potential. We did not measure DSF T_m specifically as they had to be determined experimentally, instead, we computed the intermediate structural uncertainty metrics upon AlphaFold prediction and the disorder propensity metrics imparted by a state-of-the-art method named fIDPnn (reference #35). We found that several of these features are significantly associated with protein degradation potential, and that terminal disorder propensity, or the lack of folding tightness, is a major determinant of degradation efficiency by our system.

4. Did the authors consider degrading drug-activating enzymes to demonstrate antagonism such as EthA since InhA is not susceptible? or transcription factors. It would be good to know the rationale for the choice of the 17 targets studied in this manuscript.

Reply: Thank you. We did not test drug-activating enzymes as their degradation, or the absence thereof, would only impart limited information pertaining to the degradation potential and functional vulnerability of endogenous substrates. We have now provided the rationales regarding substrate selection (Line 183-187) as well as the full- and tested list of proteins in Supplementary Tables 1 and 2, respectively. EthA was excluded from our shortlist of conserved substrates as it is not required for normal growth for neither *Mtb* nor *Msm*. In this revision, we did test two transcription factors, RbpA and NusG (Fig. 5b-c, Supplementary Fig. 12c-e, Supplementary Table 3). We found that they were both efficiently degraded by our system—probably because of their long, disordered N-termini – and that their degradation led to strong inhibition of bacterial growth.

Reviewer #3 (Remarks to the Author):

The manuscript of Won and colleagues describes the application of targeted protein degradation (TPD) in *Mycobacterium smegmatis* in an approach to examine bacterial protein proteolysis as a new therapeutic for tuberculosis. This manuscript is timely as new interventions for treating drug resistance in tuberculosis are urgently required. It follows the application of recently developed Bacterial PROTACs but utilising a chemically induced system using the small molecule rapamycin instead of specific chemical ligands used in PROTAC strategies. The work is novel and sheds light into the application of TPD in mycobacteria. The authors analysed using the system 17 essential genes and interestingly they found differences in their suitability of being targets for TPD. Overall, it is a well planned and robust study with careful experimentation and appropriate controls were used throughout. In addition to identifying targets that are suitable for PROTAC based degradation, the authors also identified that this strategy could potentiate the outcomes of current antibiotics with proteins of the same pathway. This conclusion is based on the potentiation of bedaquiline and rifampicin when AtpA and RpoA were targeted consequently. Even though this might be true for these two cases, I think the authors should have tested more drugs/gene combinations to conclude and strengthen this hypothesis further.

Reply: Thank you for your very kind comments. As we stated in our responses to all reviewers and the editor, we shared the same concern that our original reporter library was too small. To address this limitation, we have expanded our reporter testing to 72 substrates, and have evaluated the phenotypic outcomes of degrading endogenous substrates belonging to different functional pathways. We believe that our expanded dataset, along with our new machine learning prediction of protein degradation potential, have substantially strengthened our work and have imparted new insights to facilitate BacPROTAC substrate prioritization.

Minor comments:

A table with the selected genes and the rationale behind the selection would be a good addition.

Reply: Thank you for raising this issue. We have included a section in the main text to explain our target selection rationale (Lines 183-187) and provided the full- and experimentally validated list of substrates along with their protein features as Supplementary Table 1 and 2, respectively.

Typo in line 338: we were able...

Reply: Thank you for pointing it out, we have corrected the text accordingly.

More details in the method (lines 569-571) required so that it is clear how these images were captured and their subsequent analysis.

Reply: Thank you for raising this issue, we have revised the corresponding methods to cover necessary technical specifications (Lines 851-872) and have included the Python Jupyter notebooks and the associated Python scripts we used to conduct image analysis and to render microscopy images depicted in the figures.

Reviewers' Comments:

Reviewer #1:

Remarks to the Author:

The expansion of the tested degradation target proteins has substantially improved the manuscript. The authors have addressed all remaining concerns. I fully support publication of the current version of the manuscript.

Reviewer #2:

Remarks to the Author:

Outstanding manuscript, an excellent response to the prior critique. No further concerns.

Reviewer #3:

Remarks to the Author:

The revised manuscript of Won and colleagues describes the application of targeted protein degradation (TPD) in *Mycobacterium smegmatis* in an effort to test this new therapeutic modality using a genetic screening approach. The revised manuscript expands on the selection of protein targets that were selected for proteolysis and in addition, a machine learning algorithm was developed to test the suitability of bacterial PROTACs against a selection of essential mycobacterial proteins. This work is significant to the new field of bacPROTACs and will aid the discovery and development of future TPD approaches in Mycobacteria. The methodology is sound and conclusions are well supported. In addition, supplementary material as well as detailed methodology is adequate for the work to be reproducible. Overall, a very nice piece of work.